# Sound and Complete Neural Network Repair with Minimality and Locality Guarantees

**Feisi Fu**
Division of System Engineering
Boston University
`fufeisi@bu.edu`

**Wenchao Li**
Department of Electrical and Computer Engineering
Boston University
`wenchao@bu.edu`

## Abstract

We present a novel methodology for repairing neural networks that use ReLU activation functions. Unlike existing methods that rely on modifying the weights of a neural network which can induce a global change in the function space, our approach applies only a localized change in the function space while still guaranteeing the removal of the buggy behavior. By leveraging the piecewise linear nature of ReLU networks, our approach can efficiently construct a patch network tailored to the linear region where the buggy input resides, which when combined with the original network, provably corrects the behavior on the buggy input. Our method is both sound and complete – the repaired network is guaranteed to fix the buggy input, and a patch is guaranteed to be found for any buggy input. Moreover, our approach preserves the continuous piecewise linear nature of ReLU networks, automatically generalizes the repair to all the points including other undetected buggy inputs inside the repair region, is minimal in terms of changes in the function space, and guarantees that outputs on inputs away from the repair region are unaltered. On several benchmarks, we show that our approach significantly outperforms existing methods in terms of locality and limiting negative side effects.

## 1 Introduction

Deep neural networks (DNNs) have demonstrated impressive performances on a wide variety of applications ranging from transportation Bojarski et al. (2016) to health care Shahid et al. (2019). However, DNNs are not perfect. In many cases, especially when the DNNs are used in safety-critical contexts, it is important to correct erroneous outputs of a DNN as they are discovered after training. For instance, a neural network in charge of giving control advisories to the pilots in an aircraft collision avoidance system, such as the ACAS Xu network from Julian et al. (2019), may produce an incorrect advisory for certain situations and cause the aircraft to turn towards the incoming aircraft, thereby jeopardizing the safety of both airplanes. In this paper, we consider the problem of *neural network repair*, i.e. given a trained neural network and a set of buggy inputs (inputs on which the neural network produces incorrect predictions), repair the network so that the resulting network on those buggy inputs behave according to some given correctness specification. Ideally, the changes to the neural network function should be small so that the outputs on other inputs are either unchanged or altered in a small way. Existing works on neural network repair roughly fall into three categories.

1. *Retraining/fine-tuning.* The idea is to retrain or fine-tune the network with the newly identified buggy inputs and the corresponding corrected outputs. Methods include counterexample-guided data augmentation Dreossi et al. (2018); Ren et al. (2020), editable training Sinitsin et al. (2020) and training input selection Ma et al. (2018). One major weakness of these approaches is the lack of formal guarantees – at the end of retraining/fine-tuning, there is no guarantee that the given buggy inputs are fixed and no new bugs are introduced. In addition, retraining can be very expensive and requires access to the original training data which is impractical in cases where the neural network is obtained from a third party or the training data is private. Fine-tuning, on the other hand, often faces the issue of catastrophic forgetting Kirkpatrick et al. (2017).

2. *Direct weight modification.* These approaches directly manipulate the weights in a neural network to fix the buggy inputs. The repair problem is typically cast into an optimization problem

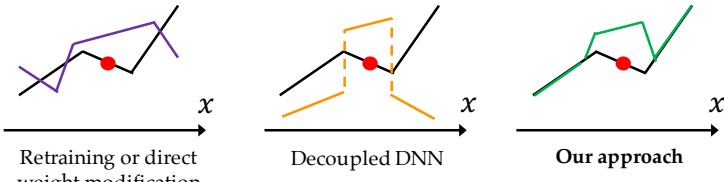

Figure 1: Comparison of different approaches to the neural network repair problem. The black lines represent the original neural network function. The red dot represents the buggy input. The colored lines represent the functions after the repairs are done.

or a verification problem. For example, Dong et al. (2020) proposes to minimize a loss defined on the buggy inputs. Goldberger et al. (2020) uses an SMT solver to identify minimal weight changes to the output layer of the network so that the undesirable behaviors are removed. Usman et al. (2021) first locates potentially faulty weights in a layer and then uses constraint solving to find a weights modification that can fix the failures. In general, the optimization-based approach cannot guarantee removal of the buggy behaviors, and the verification-based approach does not scale beyond networks of a few hundred neurons. In addition, these approaches can suffer from substantial accuracy drops on normal inputs since *weight changes may be a poor proxy for changes in the function space*.

3. *Architecture extension.* The third category of approaches extends the given NN architecture, such as by introducing more weight parameters, to facilitate more efficient repairs. The so-called Decoupled DNN architecture Sotoudeh & Thakur (2021) is the only work we know that falls into this category. Their idea is to decouple the activations of the network from values of the network by augmenting the original network. Their construction allows the formulation of any single-layer repair as an linear programming (LP) problem. The decoupling, however, causes the repaired network to become discontinuous (in the functional sense). In addition, it still cannot isolate the output change to a single buggy input from the rest of the inputs.

In addition to the aforementioned limitations, a common weakness that is shared amongst these methods is that the induced changes, as a result of either retraining or direct weight modification, are *global*. This means that a correct behavior on another input, regardless of how far it is away from the buggy input, may not be preserved by the repair. Worse still, the repair on a new buggy input can end up invalidating the repair on a previous buggy input. The fundamental issue here is that limiting the changes to a few weights or a single layer only poses a structural constraint (often for ease of computation); it does not limit the changes on the input-output mapping of the neural network. It is known that even a single weight change can have a global effect on the output of a neural network.

In this paper, we propose REASSURE, a novel methodology for neural network repair with locality, minimality, soundness and completeness guarantees. Our methodology targets continuous piecewise linear (CPWL) neural networks, specifically those that use the ReLU activation functions. The key idea of our approach is to *leverage the CPWL property of ReLU networks to synthesize a patch network tailored to the linear region where the buggy input resides, which when combined with the original network, provably corrects the behavior on the buggy input*. Our approach is both sound and complete – the repaired network is guaranteed to fix the buggy input, and a patch is guaranteed to be found for any buggy input. Moreover, our approach preserves the CPWL nature of ReLU networks, automatically generalizes the repair to all the points including other undetected buggy inputs inside the repair region, is minimal in terms of changes in the function space, and guarantees that outputs on inputs away from the repair region are unaltered. Figure 1 provides an illustrative comparison of our approach with other methods. Table 1 compares our approach with representative related works in terms of theoretical guarantees. We summarize our contributions below.

1. We present REASSURE, the first sound and complete repair methodology for ReLU networks with strong theoretical guarantees.

2. Our technique synthesizes a patch network, which when combined with the original neural network, provably corrects the behavior on the buggy input. This approach is a significant departure from existing methods that rely on retraining or direct weight manipulation.

3. Across a set of benchmarks, REASSURE can efficiently correct a set of buggy inputs or buggy areas with little or no change to the accuracy and overall functionality of the network.

|  | REASSURE | Retrain | MDNN | Editable Fine-Tuning | PRDNN |
|---|---|---|---|---|---|
| Preservation of CPWL | Yes | Yes | Yes | Yes | No |
| Soundness | Yes | No | Yes | No | Yes |
| Completeness | Yes | No | No | No | No |
| Area Repair | Yes | No | No | No | Yes |
| Minimal Change | Yes (Function Space) | No | Yes (Weight Space) | No | Yes (Weight Space) |
| Localized Change | Yes | No | No | No | No |
| Limited Side Effect | Yes | No | No | No | No |

Table 1: Comparing REASSURE with representative related works in terms of theoretical guarantees. CPWL stands for continuous piecewise linearity. Area repair means repairing all the (infinitely many) points inside an area. Limited side effect means the repair can limit potential adverse effects on other inputs. MDNN is the verification-based approach from Goldberger et al. (2020). PRDNN is the Decoupled DNN approach from Sotoudeh & Thakur (2021). REASSURE is the only method that can provide all the guarantees.

## 2 BACKGROUND

### 2.1 DEEP NEURAL NETWORKS

An $R$-layer feed-forward DNN $f = \kappa_R \circ \sigma \circ \kappa_{R-1} \circ ... \circ \sigma \circ \kappa_1 : X \to Y$ is a composition of linear functions $\kappa_r, r = 1, 2, ..., R$ and activation function $\sigma$, where $X \subseteq \mathbb{R}^m$ is a bounded input domain and $Y \subseteq \mathbb{R}^n$ is the output domain. Weights and biases of linear function $\{\kappa_r\}_{r=1,2,...,R}$ are parameters of the DNN.

We call the first $R - 1$ layers hidden layers and the $R$-th layer the output layer. We use $z_j^i$ to denote the $i$-th neuron (before activation) in the $j$-th hidden layer.

In this paper, we focus on ReLU DNNs, i.e. DNNs that use only the ReLU activation functions. It is known that an $\mathbb{R}^m \to \mathbb{R}$ function is representable by a ReLU DNN *if and only if* it is a continuous piecewise linear (CPWL) function Arora et al. (2016). The ReLU function is defined as $\sigma(x) = \max(x, 0)$. We say that $\sigma(x)$ is activated if $\sigma(x) = x$.

### 2.2 LINEAR REGIONS

A linear region $\mathcal{A}$ is the set of inputs that correspond to the same activation pattern in a ReLU DNN $f$ Serra et al. (2017). Geometrically, this corresponds to a convex polytope, which is an intersection of half spaces, in the input space $X$ on which $f$ is linear. We use $f|_{\mathcal{A}}$ to denote the part of $f$ on $\mathcal{A}$.

### 2.3 CORRECTNESS SPECIFICATION

A correctness specification $\Phi = (\Phi_{in}, \Phi_{out})$ is a tuple of two polytopes, where $\Phi_{in}$ is the union of some linear regions and $\Phi_{out}$ is a convex polytope. A DNN $f$ is said to meet a specification $\Phi = (\Phi_{in}, \Phi_{out})$, denoted as $f \models \Phi$, if and only if $\forall x \in \Phi_{in}, f(x) \in \Phi_{out}$.

**Example 1.** *For a classification problem, we can formally write the specification that "the prediction of any point in an area $\mathcal{A}$ is class $k$" as $\Phi = (\Phi_{in}, \Phi_{out})$, where $\Phi_{in} = \mathcal{A}$ and $\Phi_{out} = \{y \in \mathbb{R}^n \mid y_k \geq y_i, \forall i \neq k\}$*[1].

### 2.4 PROBLEM DEFINITION

In this paper, we consider the following two repair problems.

**Definition 1** (Area repair). *Given a correctness specification $\Phi = (\Phi_{in}, \Phi_{out})$ and a ReLU DNN $f \not\models \Phi$, the area repair problem is to find a modified ReLU DNN $\widehat{f}$ such that $\widehat{f} \models \Phi$.*

Note that we do not require $\widehat{f}$ to have the same structure or parameters as $f$ in this definition. If $\Phi_{in}$ contains a single (buggy) linear region, we refer to this as *single-region repair*. If $\Phi_{in}$ contains multiple (buggy) linear regions, we refer to it as *multi-region repair*.

---

[1]Note that here $y$ is the output of the layer right before the softmax layer in a classification network.

**Definition 2** (Point-wise repair). *Given a set of buggy inputs $\{\widetilde{x}_1, \ldots, \widetilde{x}_L\} \subset \Phi_{in}$ with their corresponding correct outputs $\{y_1, \ldots, y_L\}$ and a ReLU DNN $f$, the point-wise repair problem is to find a modified ReLU DNN $\widehat{f}$ such that $\forall i, \widehat{f}(\widetilde{x}_i) = y_i$.*

We call the minimal variants of area repair and point-wise repair *minimal area repair* and *minimal point-wise repair* respectively. Minimality here is defined with respect to the maximum distance between $f$ and $\widehat{f}$ over the whole input domain $X$. A point-wise repair can be generalized to an area repair through the following result.

## 2.5 FROM BUGGY INPUTS TO BUGGY LINEAR REGIONS

The linear region where an input $x$ resides can be computed as follows.

**Lemma 1.** *Lee et al. (2019) Consider a ReLU DNN $f$ and an input $x \in X$. For every neuron $z_j^i$, it induces a feasible set*

$$\mathcal{A}_j^i(x) = \begin{cases} \{\bar{x} \in X | (\nabla_x z_j^i)^T \bar{x} + z_j^i - (\nabla_x z_j^i)^T x \geq 0\} & \text{if } z_j^i \geq 0 \\ \{\bar{x} \in X | (\nabla_x z_j^i)^T \bar{x} + z_j^i - (\nabla_x z_j^i)^T x \leq 0\} & \text{if } z_j^i < 0 \end{cases} \tag{1}$$

*The set $\mathcal{A}(x) = \cap_{i,j} \mathcal{A}_j^i(x)$ is the linear region that includes $x$. Note that $\mathcal{A}(x)$ is essentially the H-representation of the corresponding convex polytope.*

## 2.6 REPAIR DESIDERATA

We argue that an effective repair algorithm for ReLU DNN should satisfy the following criteria.

**1. Preservation of CPWL**: Given that the original network $f$ models a CPWL function, the repaired network $\widehat{f}$ should still model a CPWL function. **2. Soundness**: A sound repair should completely remove the known buggy behaviors, i.e. it is a solution to the point-wise repair problem defined in Definition 2. **3. Completeness**: Ideally, the algorithm should always be able find a repair for any given buggy input if it exists. **4. Generalization**: If there exists another buggy input $\widetilde{x}'$ in the neighborhood of $\widetilde{x}$ (e.g. the same linear region), then the repair should also fix it. For example, suppose we have an $\widetilde{x}$ that violates a specification which requires the output to be within some range. It is almost guaranteed that there exists another (and infinitely many) $\widetilde{x}'$ in the same linear region that also violates the specification. **5. Locality**: We argue that a good repair should only induce a localized change to $f$ in the function space. For example, in the context of ReLU DNN, if a linear region $\mathcal{B}$ does not border the repair region $\mathcal{A}$, i.e. $\mathcal{B} \cap \mathcal{A} = \emptyset$, then $\widehat{f}|_{\mathcal{B}}(x) = f|_{\mathcal{B}}(x)$. **6. Minimality**: Some notion of distance between $f$ and $\widehat{f}$ such as $\max|f - \widehat{f}|$ should be minimized. Note that this is a significant departure from existing methods that focus on minimizing the change in weights which has no guarantee on the amount of change in the function space. **7. Limited side effect**: Repairing a buggy point should not adversely affect points that were originally correct. For example, repairing a buggy input $\widetilde{x}$ in region $\mathcal{A}$ should not change another region from correct to incorrect. Formally, for any linear region $\mathcal{C}$ who is a neighbor of $\mathcal{A}$, i.e. $\mathcal{C} \cap \mathcal{A} \neq \emptyset$, if $f|_{\mathcal{C}} \models \Phi$, then $\widehat{f}|_{\mathcal{C}} \models \Phi$. **8. Efficiency**: The repair algorithm should terminate in polynomial time with respect to the size of the neural network and the number of buggy inputs.

## 3 OUR APPROACH

We will first describe our approach to *single-region repair* and then present our approach to *multi-region repair* which builds on results obtained from the single-region case.

Given a linear region $\mathcal{A}$, our overarching approach is to synthesize a *patch network* $h_{\mathcal{A}}$ such that $\widehat{f} = f + h_{\mathcal{A}}$ and $\widehat{f} \models \Phi$. The patch network $h_{\mathcal{A}}$ is a combination of two sub-networks: a support network $g_{\mathcal{A}}$, which behaves like a characteristic function, to determine whether an input is in $\mathcal{A}$, and an affine patch function network $p_{\mathcal{A}}(x) = \boldsymbol{c}x + d$ to ensure $(f + p_{\mathcal{A}}) \models \Phi$ on $\mathcal{A}$.

### 3.1 RUNNING EXAMPLE

We use the following example to illustrate our idea.

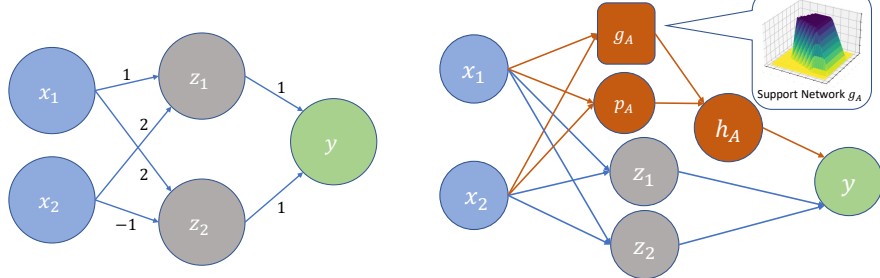

Figure 2: Left: the target DNN with buggy inputs. Right: the REASSURE-repaired DNN with the patch network shown in red. Support network $g_{\mathcal{A}}$ is for approximating the characteristic function on $\mathcal{A}$; Affine patch function $p_{\mathcal{A}}$ ensures the satisfaction of $\Phi$ on $\mathcal{A}$; The design of the patch network $h_{\mathcal{A}}$ ensures locality for the final patch.

**Example 2.** *Consider repairing the ReLU DNN $f$ in Figure 2 according to the correctness specification $\Phi : \forall x \in [0, 1]^2$, $y \in [0, 2]$. The DNN consists of a single hidden layer with two neurons $z_1$ and $z_2$, where $y = \sigma(z_1) + \sigma(z_2)$, $z_1 = x_1 + 2x_2 - 1$ and $z_2 = 2x_1 - x_2$.*

The only linear region that violates our specification is $\mathcal{A} = \{x \mid 1 \geq x_1, 1 \geq x_2, x_1 + 2x_2 - 1 \geq 0, 2x_1 - x_2 \geq 0\}$. ( $\widetilde{x} = (0.9, 0.9) \in [0, 1]^2$ but $f(\widetilde{x}) = 2.6 \notin [0, 2]$)

The network $f(x)$ on the linear region $\mathcal{A}$ is the affine function $f|_{\mathcal{A}}(x) = 3x_1 + x_2 - 1$. Our algorithm first sets up an affine function $p_{\mathcal{A}}(x)$ that minimally repairs $f$ on $\mathcal{A}$, such that $\forall x \in \mathcal{A}, f(x) + p_{\mathcal{A}}(x) \in [0, 2]$. Later in the paper, we will show $p_{\mathcal{A}}(x)$ can be found by solving a LP problem. The resulting patch function is $p_{\mathcal{A}}(x) = -\frac{1}{2}x_1 - \frac{1}{2}x_2$.

However, directly apply $f(x) + p_{\mathcal{A}}(x)$ as the patch network will have side effects on areas outside of $\mathcal{A}$. Our strategy is to combine $p_{\mathcal{A}}(x)$ with a support network $g_{\mathcal{A}}(x)$ which outputs 1 on $\mathcal{A}$ and drops to 0 quickly outside of $\mathcal{A}$. The final repaired network is $f(x) + \sigma(p_{\mathcal{A}}(x) + g_{\mathcal{A}}(x, 10) - 1) - \sigma(-p_{\mathcal{A}}(x) + g_{\mathcal{A}}(x, 10) - 1)$. This structure makes $p_{\mathcal{A}}$ almost only active on $\mathcal{A}$ and achieve a localized repair. Observe that this is still a ReLU DNN.

## 3.2 Support Networks

Support networks are neural networks with a special structure that can approximate the characteristic function of a convex polytope. They are keys to ensuring localized repairs in our algorithm.

Assume that the linear region we need to repair is $\mathcal{A} = \{x \mid a_i x \leq b_i, i \in I\}$, where $|I|$ is the number of linear inequalities. The support network of $\mathcal{A}$ is defined as:

$$g_{\mathcal{A}}(x, \gamma) = \sigma(\sum_{i \in I} g(b_i - a_i x, \gamma) - |I| + 1) \tag{2}$$

where $g(x, \gamma) = \sigma(\gamma x + 1) - \sigma(\gamma x)$ and $\gamma \in \mathbb{R}$ is a parameter of our algorithm that controls how quickly $g_{\mathcal{A}}(x, \gamma)$ goes to zero outside of $\mathcal{A}$.

**Remark**: For any $x \in \mathcal{A}$, we have $g_{\mathcal{A}}(x, \gamma) = 1$, i.e. the support network is fully activated. For any $x \notin \mathcal{A}$, if for one of $i \in I$, we have $a_i x - b_i \leq -1/\gamma$, then $g_{\mathcal{A}}(x, \gamma) = 0$.

Observe that $g_{\mathcal{A}}(x, \gamma)$ is not zero when $x$ is very close to $\mathcal{A}$ due to the requirement for preserving CPWL. In Theorem 3, we prove that we can still guarantee limited side effects on the whole input domain outside of $\mathcal{A}$ with this construction.

## 3.3 Affine Patch Functions

We consider an affine patch function $p_{\mathcal{A}}(x) = \mathbf{c}x + d$, where matrix $\mathbf{c}$ and vector $d$ are undetermined coefficients. In a later section, the design of patch network will ensure that on the patch area $\mathcal{A}$, the repaired network is $f(x) + p_{\mathcal{A}}(x)$. We will first consider finding appropriate $\mathbf{c}$ and $d$ such that $f(x) + p_{\mathcal{A}}(x)$ satisfy the specification on $\mathcal{A}$.

To satisfy the specification $\Phi$, we need $f(x) + p_{\mathcal{A}}(x) \in \Phi_{out}$ for all $x \in \mathcal{A}$. To obtain a minimal repair, we minimize $\max_{x \in \mathcal{A}} |p_{\mathcal{A}}(x)|$. Thus, we can formulate the following optimization problem

$$\begin{cases} \min_{\boldsymbol{c},d} \max_{x \in \mathcal{A}} |p_{\mathcal{A}}(x)| = |\boldsymbol{c}x + d| \\ (\boldsymbol{c}, d) \in \{(\boldsymbol{c}, d) \mid f(x) + \boldsymbol{c}x + d \in \Phi_{out}, \forall x \in \mathcal{A}\} \end{cases} \tag{3}$$

Notice that this is not an LP since both $\boldsymbol{c}$ and $x$ are variables and we have a $\boldsymbol{c}x$ term in the objective.

In general, one can solve it by enumerating all the vertices of $\mathcal{A}$. Suppose that $\{v_s | s = 1, 2, ..., S\}$ is the set of vertices of $\mathcal{A}$. Since $\Phi_{out}$ is a convex polytope, we have

$$f(x) + p_{\mathcal{A}}(x) \in \Phi_{out} \text{ for all } x \in \mathcal{A} \Leftrightarrow f(v_s) + p_{\mathcal{A}}(v_s) \in \Phi_{out} \text{ for } s = 1, 2, ..., S. \tag{4}$$

and

$$\max_{x \in \mathcal{A}} |\boldsymbol{c}x + d| = \max_{s=1,2,...,S} |cv_s + d| \tag{5}$$

Hence, we can solve the following equivalent LP.

$$\begin{cases} \min_{\boldsymbol{c},d} H \\ H \geq (cv_s + d)_i, H \geq -(cv_s + d)_i, \text{for } s = 1, 2, ..., S \text{ and } i = 1, 2, ..., m \\ f(v_s) + p_{\mathcal{A}}(v_s) \in \Phi_{out}, \text{for } s = 1, 2, ..., S \end{cases} \tag{6}$$

where $H \in \mathbb{R}$ and will take $\max_{s=1,2,...,S} |cv_s + d|$ when optimal.

In general, the number of vertices of a convex polytope can be exponential in the size of its H-representation Henk et al. (1997) and enumerating the vertices of a convex polytope is known to be expensive especially when the input dimension is large Bremner (1997). In Appendix 7.1, we show that we can solve programming 3 via LP *without vertex enumeration* for many useful cases such as the classification problem in Example 1 and make our algorithm much more efficient.

## 3.4 SINGLE-REGION REPAIRS

With a support network $g_{\mathcal{A}}$ and an affine patch function $p_{\mathcal{A}}$, we can synthesize the final patch network as follows:

$$h_{\mathcal{A}}(x, \gamma) = \sigma(p_{\mathcal{A}}(x) + K \cdot g_{\mathcal{A}}(x, \gamma) - K) - \sigma(-p_{\mathcal{A}}(x) + K \cdot g_{\mathcal{A}}(x, \gamma) - K) \tag{7}$$

where $K$ is a vector where every entry is equal to the upper bound of $\{|p_{\mathcal{A}}(x)|_{+\infty} | x \in X\}$.

**Remark:** For $x \in \mathcal{A}$, $g_{\mathcal{A}}(x, \gamma) = 1$, then we have $h_{\mathcal{A}}(x, \gamma) = \sigma(p_{\mathcal{A}}(x)) - \sigma(-p_{\mathcal{A}}(x)) = p_{\mathcal{A}}(x)$. For $x \notin \mathcal{A}$, $g_{\mathcal{A}}(x, \gamma)$ goes to zero quickly if $\gamma$ is large. When $g_{\mathcal{A}}(x, \gamma) = 0$, we have $h_{\mathcal{A}}(x, \gamma) = \sigma(p_{\mathcal{A}}(x) - K) - \sigma(-p_{\mathcal{A}}(x) - K) = 0$.

The repaired network $\widehat{f}(x) = f(x) + h_{\mathcal{A}}(x, \gamma)$. Since $f$ and $h_{\mathcal{A}}$ are both ReLU DNNs, we have $\widehat{f}$ is also a ReLU DNN. We will give the formal guarantees on correctness in Theorem 1.

## 3.5 MULTI-REGION REPAIRS

Suppose there are two linear regions, $\mathcal{A}_1$ and $\mathcal{A}_2$, that need to be repaired, and we have generated the affine patch function $p_{\mathcal{A}_1}(x)$ for $\mathcal{A}_1$ and $p_{\mathcal{A}_2}(x)$ for $\mathcal{A}_2$.

If $\mathcal{A}_1 \cap \mathcal{A}_2 = \emptyset$, then we can repair $f(x)$ with $\widehat{f}(x) = f(x) + h_{\mathcal{A}_1}(x, \gamma) + h_{\mathcal{A}_2}(x, \gamma)$ directly, since $h_{\mathcal{A}_1}(x, \gamma)$ and $h_{\mathcal{A}_2}(x, \gamma)$ will not be nonzero at the same time when $\gamma$ is large enough.

However, if $\mathcal{A}_1 \cap \mathcal{A}_2 \neq \emptyset$, for any $x \in \mathcal{A}_1 \cap \mathcal{A}_2$, both $h_{\mathcal{A}_1}(x, \gamma)$ and $h_{\mathcal{A}_2}(x, \gamma)$ will alter the value of $f$ on $x$, which will invalidate both repairs and cannot guarantee that the repaired DNN will meet the specification $\Phi$. To avoid such over-repairs, our strategy is to first repair $\mathcal{A}_1 \cup \mathcal{A}_2$ with $p_{\mathcal{A}_1}(x)$, and then repair $\mathcal{A}_2$ with $p_{\mathcal{A}_2}(x) - p_{\mathcal{A}_1}(x)$. Figure 3 provides an illustration of a three-region case.

In general, for multi-region repair, we note $\{\mathcal{A}_l\}_{l=1,2,...,L}$ are all the buggy linear regions. Then we compute the support network $g_{\mathcal{A}_l}(x, \gamma)$ and affine patch function $p_{\mathcal{A}_l}(x)$ for each $\mathcal{A}_l$. Note that this computation can be done in parallel.

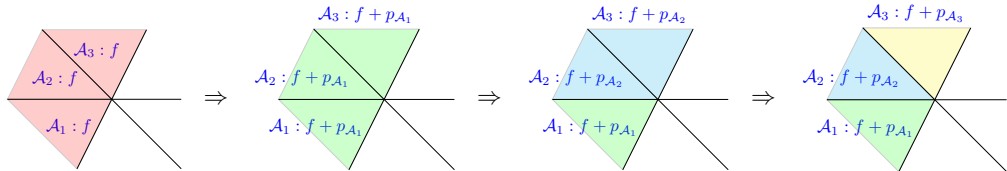

Figure 3: An illustration of multi-region repair with three different repair regions. Left: the original DNN; Middle Left: repair $\mathcal{A}_1 \cup \mathcal{A}_2 \cup \mathcal{A}_3$ with $p_{\mathcal{A}_1}$; Middle Right: repair $\mathcal{A}_2 \cup \mathcal{A}_3$ with $p_{\mathcal{A}_2} - p_{\mathcal{A}_1}$; Right: repair $\mathcal{A}_3$ with $p_{\mathcal{A}_3} - p_{\mathcal{A}_2}$

Once we have $g_{\mathcal{A}_l}(x, \gamma)$ and $p_{\mathcal{A}_l}(x)$, we can "stitch" multiple local patches into a final patch as follows.

$$h(x, \gamma) = \sum_l [\sigma(p_{\mathcal{A}_l}(x) - p_{\mathcal{A}_{l-1}}(x) + \max_{j \geq l}\{g_{\mathcal{A}_j}(x, \gamma)\}K_l - K_l)$$
$$-\sigma(-p_{\mathcal{A}_l}(x) + p_{\mathcal{A}_{l-1}}(x) + \max_{j \geq l}\{g_{\mathcal{A}_j}(x, \gamma)\}K_l - K_l)] \tag{8}$$

where $K_l$ is the upper bound of $\{|p_{\mathcal{A}_l}(x) - p_{\mathcal{A}_{l-1}}(x)|_\infty | x \in X\}$ and $p_{\mathcal{A}_0}(x) = 0$.

**Remark**: $\max_{j \geq l}\{g_{\mathcal{A}_j}(x, \gamma)\}$ is a support function for $\cup_{j \geq l}\mathcal{A}_j$ and its value is 1 for any $x \in \cup_{j \geq l}\mathcal{A}_j$.

## 4 THEORETICAL GUARANTEES

In this section, we present the theoretical guarantees that REASSURE provides, and point the readers to proofs of the theorems in the Appendix.

**Theorem 1** (Soundness). *The repaired DNN $\widehat{f}$ returned by REASSURE is guaranteed to satisfy the specification $\Phi$.*

**Theorem 2** (Completeness). *REASSURE can always find a solution to the minimal point-wise repair or the minimal area repair problem.*

For any $\mathcal{A}$, the support network ensures that the patch network goes to zero quickly when $x$ is away from $\mathcal{A}$. However, it still makes a small change on the neighbors of $\mathcal{A}$. The following theorem shows that for a big enough $\gamma$, the patch network would not change a correct region into incorrect.

**Theorem 3** (Limited Side Effect). *Given a correctness property $\Phi = (\Phi_{in}, \Phi_{out})$, a patch region $\mathcal{A}$ and the corresponding patch network $h(x, \gamma)$, there exists a positive number $\Gamma$ such that for any $\gamma \geq \Gamma$, we have*

    *1. for any linear region $\mathcal{B}$, if $\mathcal{B} \cap \mathcal{A} = \emptyset$, then $\widehat{f}(x, \gamma) = f(x)$;*

    *2. for any linear region $\mathcal{C}$ who is a neighbor of $\mathcal{A}$ ($\mathcal{C} \cap \mathcal{A} \neq \emptyset$), if $f|_\mathcal{C} \models \Phi$, then $\widehat{f}_\mathcal{C}(x, \gamma) \models \Phi$.*

**Corollary 1** (Incremental Repair). *For multiple-region repair, the patch for a new region $\mathcal{A}'$ would not cause a previous patched region $\mathcal{A}$ to become incorrect.*

**Theorem 4** (Minimum Repair). *For any ReLU DNN $\tilde{f}$, which is linear on a patch region $\mathcal{A}$ and satisfies the specification $\Phi$, there exists a positive number $\Gamma$, such that for all $\gamma \geq \Gamma$,*

$$\max_{x \in X} |\tilde{f}(x) - f(x)| \geq \max_{x \in X} |h_\mathcal{A}(x, \gamma)|. \tag{9}$$

**Theorem 5** (Polynomial-Time Efficiency). *REASSURE terminates in polynomial-time in the size of the neural network and the number of buggy linear regions when $\Phi_{out}$ takes the form of $\{y \mid q_l \leq Py \leq q_u\}$ where $P$ is a full row rank matrix and $-\infty \leq q_l[i] \leq q_u[i] \leq +\infty$ ($q_l[i]$ and $q_u[i]$ are the i-th elements of $q_l$ and $q_u$ respectively).*

## 5 EXPERIMENTS

In this Section, we compare REASSURE with state-of-the-art methods on both *point-wise repairs* and *area repairs*. The experiments were designed to answer the following questions: **(Effectiveness)**

| | | REASSURE | | | | | Retrain *(Requires Training Data)* | | | |
|---|---|---|---|---|---|---|---|---|---|---|
| #P | $ND(L_\infty)$ | $ND(L_2)$ | $NDP(L_\infty)$ | $NDP(L_2)$ | Acc | $ND(L_\infty)$ | $ND(L_2)$ | $NDP(L_\infty)$ | $NDP(L_2)$ | Acc |
| 10 | **0.01%** | **0.01%** | **25.75%** | **24.78%** | **98.1%** | 1.13% | 1.09% | 77.86% | 77.60% | **98.1%** |
| 20 | **0.03%** | **0.02%** | **19.17%** | **18.70%** | 98.2% | 0.92% | 0.89% | 77.14% | 76.29% | **98.4%** |
| 50 | **0.06%** | **0.06%** | **24.64%** | **23.79%** | 98.5% | 0.84% | 0.82% | 84.17% | 82.15% | **98.7%** |
| 100 | **0.11%** | **0.12%** | **25.40%** | **24.44%** | 99.0% | 0.84% | 0.82% | 84.83% | 83.05% | **99.0%** |

| | | Fine-Tuning | | | | | PRDNN | | | |
|---|---|---|---|---|---|---|---|---|---|---|
| #P | $ND(L_\infty)$ | $ND(L_2)$ | $NDP(L_\infty)$ | $NDP(L_2)$ | Acc | $ND(L_\infty)$ | $ND(L_2)$ | $NDP(L_\infty)$ | $NDP(L_2)$ | Acc |
| 10 | 2.20% | 2.11% | 67.61% | 65.52% | 97.6% | 1.41% | 1.34% | 34.60% | 33.59% | 97.8% |
| 20 | 23.19% | 22.35% | 82.87% | 78.55% | 78.6% | 2.88% | 2.74% | 43.63% | 41.78% | 97.1% |
| 50 | 35.78% | 34.04% | 84.73% | 80.58% | 67.0% | 4.79% | 4.47% | 49.37% | 46.31% | 96.7% |
| 100 | 23.83% | 22.11% | 79.73% | 76.57% | 81.9% | 9.16% | 8.20% | 51.23% | 46.34% | 96.1% |

Table 2: Point-wise Repairs on MNIST. We use the first hidden layer as the repair layer for PRDNN. The test accuracy of the original DNN is 98.0%. #P: number of buggy points to repair. $ND(L_\infty)$, $ND(L_2)$: average ($L_\infty$, $L_2$) norm difference on both training and test data. $NDP(L_\infty)$, $NDP(L_2)$: average ($L_\infty$, $L_2$) norm difference on random sampled points near the buggy points. Acc: accuracy on test data. *Note that* REASSURE *automatically performs area repairs on 784-dimensional inputs.*

How effective is a repair in removing known buggy behaviors? **(Locality)** How much side effect (i.e. modification outside the patch area in the function space) does a repair produce? **(Function Change)** How much does a repair change the original neural network in the function space? **(Performance)** Whether and how much does a repair adversely affect the overall performance of the neural network?

We consider the following **evaluation criteria**: 1. **Efficacy (E)**: % of given buggy points or buggy linear regions that are repaired. 2. **Norm Difference (ND)**: average normalized norm ($L_\infty$ or $L_2$) difference between the original DNN and the repaired DNN on a set of inputs (e.g. training and testing data; more details in the tables). We use ND to measure how a repair *change* the original neural network on *function space*. 3. **Norm Difference on Patch Area (NDP)**: average normalized norm ($L_\infty$ or $L_2$) difference between the original DNN and the repaired DNN on patch areas (calculated on random sampled points on patch areas or near the buggy points; details in the tables). We use NDP to measure the *locality* of a repair. 4. **Accuracy (Acc)**: accuracy on training or testing data to measure the extent to which a repair preserves the performance of the original neural network. 5. **Negative Side Effect (NSE)**: NSE is only for area repair. It is the percentage of correct linear regions (outside of patch area) that become incorrect after a repair. If a repair has a nonzero NSE, the new repair may invalidate a previous repair and lead to a circular repair problem.

We compared REASSURE with the representative related works in Table 1. REASSURE, MDNN and PRDNN guarantee to repair all the buggy points (linear regions). Retrain and Fine-Tuning cannot guarantee 100% efficacy in general and we run them until all the buggy points are repaired.

## 5.1 Point-wise Repairs: MNIST

We train a ReLU DNN on the MNIST dataset LeCun (1998) as the target DNN. The goal of a repair is to fix the behaviors of the target DNN on buggy inputs that are found in the test dataset. Thus, the repaired DNN is expected to produce correct predictions for all the buggy inputs.

The results are shown in Table 2. REASSURE achieves almost zero modification outside the patch area (ND) amongst all four methods. In addition, REASSURE produces the smallest modification on the patch area (NDP) and preserves the performance of the original DNN (almost no drop on Acc).

## 5.2 Area Repairs: HCAS

To the best of our knowledge, Sotoudeh & Thakur (2021) is the only other method that supports area repairs. In this experiment, we compare REASSURE with Sotoudeh & Thakur (2021) on an experiment where the setting is similar to the *2D Polytope ACAS Xu repair* in their paper.

Sotoudeh & Thakur (2021) does not include a vertex enumeration tool (which is required for setting up their LP problem) in their code. We use pycddlib Troffaes (2018) to perform the vertex enumeration step when evaluating PRDNN. Note that the vertex enumeration tool does not affect the experimental results except running time.

| | | REASSURE | | | | PRDNN | | | |
|---|---|---|---|---|---|---|---|---|---|
| #A | ND($L_\infty$) | NDP($L_\infty$) | NSE | Acc | T | ND($L_\infty$) | NDP($L_\infty$) | NSE | Acc | T |
| 10 | **0.00%** | **0.0%** | **0%** | **98.1%** | **1.0422** | 0.10% | 31.6% | 4% | 89.6% | 2.90+0.100 |
| 20 | **0.00%** | **2.2%** | **0%** | **98.1%** | **1.1856** | 0.15% | 37.2% | 8% | 83.1% | 5.81+0.185 |
| 50 | **0.00%** | **17.6%** | **0%** | **98.1%** | **1.8174** | 0.15% | 38.4% | 8% | 83.8% | 14.54+0.388 |
| 87 | **0.04%** | **45.9%** | **0%** | **97.8%** | **2.4571** | 0.14% | 46.6% | **0%** | 85.6% | 25.30+0.714 |

Table 3: Area Repairs on HCAS. We use the the first hidden layer as the repair layer for PRDNN. Results on PRDNN using the last layer (which are inferior to using the first layer) are shown in Table 6 in the Appendix 7.4. The test accuracy of the original DNN is $97.9\%$. #A: number of buggy linear regions to repair. ND($L_\infty$): average $L_\infty$ norm difference on training data. NDP($L_\infty$): average $L_\infty$ norm difference on random sampled data on input constraints of specification 1. NSE: $\%$ of correct linear regions changed to incorrect by the repair. Acc: accuracy on training data (no testing data available). T: running time in seconds. For PRDNN, the first running time is for enumerating all the vertices of the polytopes and the second is for solving the LP problem in PRDNN.

| | REASSURE *(feature space)* | | | Retrain *(Requires Training Data)* | | | Fine-Tuning | | | PRDNN | | |
|---|---|---|---|---|---|---|---|---|---|---|---|---|
| #P | ND($L_\infty$) | ND($L_2$) | Acc | ND($L_\infty$) | ND($L_2$) | Acc | ND($L_\infty$) | ND($L_2$) | Acc | ND($L_\infty$) | ND($L_2$) | Acc |
| 10 | **0.15%** | **0.13%** | **82.5%** | 43.43% | 36.04% | 80.1% | 29.45% | 25.27% | 77.9% | 22.93% | 21.13% | 82.1% |
| 20 | **0.12%** | **0.11%** | 81.3% | 42.78% | 35.69% | **82.9%** | 69.16% | 57.47% | 68.5% | 21.91% | 20.03% | 80.1% |
| 50 | **0.79%** | **0.70%** | 81.3% | 50.23%* | 42.86%* | **82.1%*** | 76.69% | 63.46% | 66.9% | 30.96% | 26.94% | 68.9% |

Table 4: Point-wise Repairs on ImageNet. PRDNN uses parameters in the last layer for repair. The test accuracy for the original DNN is $83.1\%$. #P: number of buggy points to repair. ND($L_\infty$), ND($L_2$): average ($L_\infty$, $L_2$) norm difference on validation data. Acc: accuracy on validation data. * means Retrain only repair 96% buggy points in 100 epochs.

We consider an area repair where the target DNN is the HCAS network (simplified version of ACAS Xu)[2] $N_{1,4}$ (previous advisory equal to 1 and time to loss of vertical separation equal to $20s$) from Julian & Kochenderfer (2019). We use Specification 1 (details in Appendix 7.4), which is similar to Property 5 in Katz et al. (2017). We compute all the linear regions for $N_{1,4}$ in the area $\Phi_{in}$ of Specification 1 and 87 buggy linear regions were found. We apply both REASSURE and PRDNN to repair those buggy linear regions. We use Specification 2 (details in Appendix 7.4), the dual of Specification 1, to test the *negative side effect (NSE)* of a repair.

The results are shown in Table 3. Both REASSURE and PRDNN successfully repair all the buggy linear regions. REASSURE produces repairs that are significantly better in terms of *locality* (ND), *minimality* (NDP) and *performance preservation* (Acc).

## 5.3 FEATURE-SPACE REPAIRS

In general, when repairing a large DNN with a high input dimension, the number of linear constraints for one patch area $\mathcal{A}$ will be huge and pose a challenge to solving the resulting LP.

One advantage of our approach, which can be used to mitigate this problem, is that it allows for point-wise and area repairs in the feature space in a principled manner, i.e. constructing a patch network starting from an intermediate layer. This approach still preserves soundness and completeness, and is fundamentally different from just picking a single layer for repair in PRDNN or MDNN. Experimental results on repairing AlexNet Krizhevsky et al. (2012) for the ImageNet dataset Russakovsky et al. (2015) in Appendix 7.4 show REASSURE *(feature space)* is still significantly better in term of *locality*(ND) and *minimality* (NDP).

## 6 CONCLUSION

We have presented a novel approach for repairing ReLU DNNs with strong theoretical guarantees. Across a set of benchmarks, our approach significantly outperforms existing methods in terms of efficacy, locality, and limiting negative side effects. Future directions include further investigation on feature-space repairs and identifying a lower-bound for $\gamma$.

---

[2]The technique in PRDNN for computing linear regions does not scale beyond two dimensions as stated in their paper. The input space of HCAS is 3D and that of ACAS Xu is 5D so we use HCAS in order to run their tool in our evaluation of area repairs.

## ACKNOWLEDGEMENT

This effort was partially supported by the Intelligence Advanced Research Projects Agency (IARPA) under the contract W911NF20C0038. The content of this paper does not necessarily reflect the position or the policy of the Government, and no official endorsement should be inferred.

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

# 7 APPENDIX

## 7.1 REPAIR VIA LINEAR PROGRAMMING

We consider the case $\Phi_{out}$ can be expressed as $\{y \,|\, q_l \leq Py \leq q_u\}$ where $P$ is a *full row rank* matrix and $-\infty \leq q_l[i] \leq q_u[i] \leq +\infty$ ($q_l[i]$ and $q_u[i]$ are the $i$-th elements of $q_l$ and $q_u$ respectively).

Consider the following optimization problem.

$$\begin{cases} \min_T \max_{x \in \mathcal{A}} |T(f(x) - f(x)| \\ q_l \leq P(T(f(x))) \leq q_u, \forall x \in \mathcal{A} \end{cases} \quad (10)$$

where $T : \mathbb{R}^n \to \mathbb{R}^n$ is a linear transformation on the DNN's output space $\mathbb{R}^n$.

**Theorem 6.** *On linear region $\mathcal{A}$, we have $f|_\mathcal{A}(x) = \boldsymbol{f_1}x + f_2$ for some matrix $\boldsymbol{f_1}$ and vector $f_2$. Assuming that $\boldsymbol{f_1}$ is full rank[3], the optimization problem in (3) and the optimization problem in (10) are equivalent.*

Note that $q_l \leq P(T(f(x))) \leq q_u$ can be achieved row by row. Thus, we can find a one-dimensional linear transformation via LPs and combine them into a single linear transformation to solve optimization problem 10.

For every row of $P$, say the $i$-th row, we can check the lower bound and upper bound of $\{P(f(x)) \,|\, \forall x \in \mathcal{A}\}$ on the $i$-th dimension by solving the following LP problems

$$lb[i] = \min_{x \in \mathcal{A}} P[i](f(x)) \quad ub[i] = \max_{x \in \mathcal{A}} P[i](f(x)) \quad (11)$$

where $P[i]$ is the $i$-th row of $P$.

Then for each row $i$, we take a minimal linear transformation $V[i](x) = v_1[i](x) + v_2[i]$ to transfer interval $[lb[i], ub[i]]$ inside interval $[q_l[i], q_u[i]]$. We can take $v_1[i] = 1$ if $q_u[i] - q_l[i] > ub[i] - lb[i]$, else $v_1[i] = \frac{q_u[i] - q_l[i]}{ub[i] - lb[i]}$. And $v_2[i] = q_l[i] - v_1[i]lb[i]$ if $|q_l[i] - v_1[i]lb[i]| \leq |v_1[i]ub[i] - q_u[i]|$, else $v_1[i]ub[i] - q_u[i]$.

Since matrix $P$ is full row rank, we can find a linear transformation $T$ that is equivalent to $V$:

$$T = \hat{P}^{-1}V\hat{P} \Rightarrow P(T(f(x))) = V(P(f(x))) \quad (12)$$

where $\hat{P} = \begin{bmatrix} P \\ P^\perp \end{bmatrix}$ is an orthogonal extension of $P$ ($P$ and $P^\perp$ are orthogonal to each other and $\hat{P}$ is a full rank square matrix).

Once we have $T$, we can obtain an affine patch function $p_\mathcal{A}(x) = T(f(x)) - f(x)$.

## 7.2 THE REASSURE ALGORITHM

---
**Algorithm 1** REASSURE
---
**Input**: A specification $\Phi = (\Phi_{in}, \Phi_{out})$, a ReLU DNN $f$ and a set of buggy points $\{\widetilde{x}_1, \ldots, \widetilde{x}_L\} \subset \Phi_{in}$.
**Output**: A repaired ReLU DNN $\widehat{f}$.
1: **for** $l = 1$ to $L$ **do**
2:      Generate the patch area $\mathcal{A}_l$ from buggy point $\widetilde{x}_l$ according to Equation (1);
3:      Generate a support network $g_\mathcal{A}$ according to Equation (2);
4:      Solve the linear programming problem (6) to find the optimal affine patch network $p_\mathcal{A}$.
5: **end for**
6: Combine all support networks $g_{\mathcal{A}_l}$ and the corresponding patch networks $p_{\mathcal{A}_l}$ to get the overall patch network $h$ according to Equation (8).
7: **return** $\widehat{f} = f + h$
---

---

[3]Note that for neural networks that are trained by a stochastic method, with probability one $\boldsymbol{f_1}$ is full rank.

### 7.3 PROOFS OF THEOREMS

We prove Theorem 1 after Corollary 1, since the proof of Theorem 1 uses the result of Corollary 1.

**Theorem 6.** *On linear region $\mathcal{A}$, we have $f|_{\mathcal{A}}(x) = \boldsymbol{f_1}x + f_2$ for some matrix $\boldsymbol{f_1}$ and vector $f_2$. Assuming that $\boldsymbol{f_1}$ is full rank[4], the optimization problem in (3) and the optimization problem in (10) are equivalent.*

*Proof.* On one side, for any $\boldsymbol{c}, d$, since $\boldsymbol{f_1}$ is full rank, there exists a linear transformation $T$, such that $T(f(x)) = T(\boldsymbol{f_1}x + f_2) = (\boldsymbol{f_1} + \boldsymbol{c})x + (f_2 + d) = f(x) + \boldsymbol{c}x + d$.

On the other side, for any $T$, since $T(f(x)) - f(x)$ is linear, there exist $\boldsymbol{c}, d$, such that $\boldsymbol{c}x + d = T(f(x)) - f(x)$. □

**Lemma 2.** *The repaired DNN $\widehat{f}$ returned by* REASSURE *is guaranteed to satisfy the specification $\Phi$ on patch area $\mathcal{A}$ in single-region repair case.*

*Proof.* By the definition of $p_{\mathcal{A}}$, we have $f(x) + p_{\mathcal{A}}(x) \in \Phi_{out}$ for all $x \in \mathcal{A}$.

For any $x \in \mathcal{A}$, we have $g_{\mathcal{A}}(x, \gamma) = 1$ and $h_{\mathcal{A}}(x, \gamma) = \sigma(p_{\mathcal{A}}(x)) - \sigma(-p_{\mathcal{A}}(x)) = p_{\mathcal{A}}(x)$. Therefore,

$$\widehat{f}(x) = f(x) + h_{\mathcal{A}}(x, \gamma) = f(x) + p_{\mathcal{A}}(x) \in \Phi_{out} \tag{13}$$

Thus, the patched neural network $\widehat{f}$ meets the specification $\Phi$ on $\mathcal{A}$. □

**Theorem 2** (Completeness). REASSURE *can always find a solution to the minimal point-wise repair or the minimal area repair problem.*

*Proof.* For every patch area $\mathcal{A}$, we can always find a support network $g_{\mathcal{A}}$. For any $\Phi_{out}$ and $\mathcal{A}$, there exists an affine function $p_{\mathcal{A}}$ such that $p_{\mathcal{A}}(x) \in \Phi_{out}, \forall x \in \mathcal{A}$. Therefore, the LP (6) is always feasible and REASSURE can find an affine patch function $p_{\mathcal{A}}$.

Once we have $g_{\mathcal{A}}$ and $p_{\mathcal{A}}$ for patch area $\mathcal{A}$, REASSURE returns a patch network either by Equation (7) or by Equation (8). □

**Theorem 3** (Limited Side Effect). *Given a correctness property $\Phi = (\Phi_{in}, \Phi_{out})$, a patch region $\mathcal{A}$ and the corresponding patch network $h(x, \gamma)$, there exists a positive number $\Gamma$ such that for any $\gamma \geq \Gamma$, we have*

*1. for any linear region $\mathcal{B}$, if $\mathcal{B} \cap \mathcal{A} = \emptyset$, then $\widehat{f}(x, \gamma) = f(x)$;*

*2. for any linear region $\mathcal{C}$ who is a neighbor of $\mathcal{A}$ ($\mathcal{C} \cap \mathcal{A} \neq \emptyset$), if $f|_{\mathcal{C}} \models \Phi$, then $\widehat{f_{\mathcal{C}}}(x, \gamma) \models \Phi$.*

*Proof.* Since a multi-region repair is a composition of multiple singe-region repairs according to Equation (8), we can prove the limited side effect of a multi-region repair by proving the limited side effect of its constituent singe-region repairs. Below, we prove the limited side effect of a singe-region repair.

Consider patch area $\mathcal{A} = \{x \mid a_i x \leq b_i, i \in I\}$ and $\mathcal{A}_{>0}(\gamma) = \{x \mid h(x, \gamma) > 0\}$.

1. Since the number of neighbors for $\mathcal{A}$ are finite, we can take a big enough $\gamma$, such that for any $\mathcal{B}$, if $\mathcal{B} \cap \mathcal{A} = \emptyset, \mathcal{B} \cap \mathcal{A}_{>0}(\gamma) = \emptyset$. Thus, we have $\widehat{f}(x, \gamma) = f(x)$ on $\mathcal{B}$.

2. For any linear region $\mathcal{C}$ who is a neighbor of $\mathcal{A}$, i.e. $\mathcal{C} \neq \mathcal{A}$ and $\mathcal{C} \cap \mathcal{A} \neq \emptyset$, $\widehat{f}$ is no longer a linear function on $\mathcal{C}$, since there are some hyperplanes introduced by our repair that will divide $\mathcal{C}$ into multiple linear regions.

Specifically, those hyperplanes are $\{x \mid \gamma(a_i x - b_i) + 1 = 0\}$ for $i \in I$, $\{x \mid \sum_{i \in I} g(a_i x - b_i, \gamma) - |I| + 1 = 0\}$, $\{x \mid p(x) + K \cdot g_{\mathcal{A}}(x, \gamma) - K = 0\}$ and $\{x \mid -p(x) + K \cdot g_{\mathcal{A}}(x, \gamma) - K = 0\}$.

For any point $x$ in those hyperplanes, it will fall into one of the following four cases.

---

[4]Note that for neural networks that are trained by a stochastic method, with probability one $\boldsymbol{f_1}$ is full rank.

(a) $x \in \{x \mid \gamma(a_i x - b_i) + 1 = 0\}$ for some $i \in I$, then $g_{\mathcal{A}}(x, \gamma) = 0$, $h(x, \gamma) = 0$ and $\widehat{f}(x) \in \Phi_{out}$;

(b) $x \in \{x \mid \sum_{i \in I} g(a_i x - b_i, \gamma) - |I| + 1 = 0\}$, then $g_{\mathcal{A}}(x, \gamma) = 0$, $h(x, \gamma) = 0$ and $\widehat{f}(x) \in \Phi_{out}$;

(c) $x \in \{x \mid p(x) + K \cdot g_{\mathcal{A}}(x, \gamma) - K = 0\}$, then $p(x) = K - K \cdot g_{\mathcal{A}}(x, \gamma) \geq 0$, $-p(x) + K \cdot g_{\mathcal{A}}(x, \gamma) - K \leq 0$, $h(x, \gamma) = 0$ and $\widehat{f}(x) \in \Phi_{out}$;

(d) $x \in \{x \mid -p(x) + K \cdot g_{\mathcal{A}}(x, \gamma) - K\}$, then $p(x) = K \cdot g_{\mathcal{A}}(x, \gamma) - K \leq 0$, $p(x) + K \cdot g_{\mathcal{A}}(x, \gamma) - K \leq 0$, $h(x, \gamma) = 0$ and $\widehat{f}(x) \in \Phi_{out}$;

By the above analysis, we have $\widehat{f}(x) \in \Phi_{out}$ for the boundary of the new linear regions. Since $\widehat{f}$ is linear on the new linear regions and $\Phi_{out}$ is convex, $\widehat{f}(x) \in \Phi_{out}$ for any $x \in \mathcal{C}$. □

**Remark**: By Theorem 3, we have that a patch would not change a correct linear region to an incorrect one.

**Corollary 1** (Incremental Repair). *For multiple-region repair, the patch for a new region $\mathcal{A}'$ would not cause a previous patched region $\mathcal{A}$ to become incorrect.*

*Proof.* After applying the patch to linear region $\mathcal{A}$, we have that the resulting network is correct on $\mathcal{A}$. When applying a new patch to another linear region $\mathcal{A}'$, by Theorem 3, the new patch would not make a correct linear region $\mathcal{A}$ incorrect. □

**Theorem 1** (Soundness). *The repaired DNN $\widehat{f}$ returned by* REASSURE *is guaranteed to satisfy the specification $\Phi$.*

*Proof.* The proof has two parts:

1. to show that $\widehat{f}$ satisfies the specification $\Phi$ on $\mathcal{A}$, and

2. to show that $\widehat{f}$ satisfies the specification $\Phi$ outside of $\mathcal{A}$.

Part 1:

Lemma (2) shows $\widehat{f}$ satisfy the specification $\Phi$ for single-region repair on $\mathcal{A}$.

For the multi-region case, consider a set of buggy linear regions $\cup_{1 \leq l \leq I} \mathcal{A}_l$ with the corresponding support neural network $g_{\mathcal{A}_l}$ and affine patch function $p_{\mathcal{A}_l}$ for each $\overline{\mathcal{A}_l}$. For the multi-region repair construction in Equation (8), we refer to $\sigma(p_{\mathcal{A}_j} - p_{\mathcal{A}_{j-1}} + \max_{k \geq j}\{g_{\mathcal{A}_k}\} K_j - K_j)$ as the $j$-th patch and $\widehat{f}_j = f + \sum_{j' \leq j} \sigma(p_{\mathcal{A}_{j'}} - p_{\mathcal{A}_{j'-1}} + \max_{k \geq j'}\{g_{\mathcal{A}_k}\} K_j - K_j)$ as the network after the $j$-th patch.

For any $x$ in patch area $\cup_{1 \leq l \leq I} \mathcal{A}_l$, we can find a $j$ such that $x \in \mathcal{A}_j$ but $x \notin \mathcal{A}_k$ for all $k > j$. After the first $j$ patches $\sigma(p_{\mathcal{A}_1}(x) + \max_{k \geq 1}\{g_{\mathcal{A}_k}(x, \gamma)\} K_1 - K_1)$, $\sigma(p_{\mathcal{A}_2}(x) - p_{\mathcal{A}_1}(x) + \max_{k \geq 2}\{g_{\mathcal{A}_k}(x, \gamma)\} K_2 - K_2)$, ... , $\sigma(p_{\mathcal{A}_j}(x) - p_{\mathcal{A}_{j-1}}(x) + \max_{k \geq j}\{g_{\mathcal{A}_k}(x, \gamma)\} K_j - K_j)$, the DNN's output at $x$ becomes $\widehat{f}_j(x) = f(x) + p_{\mathcal{A}_j}(x)$ which meets our specification $\Phi$ at $x$ by the definition of $p_{\mathcal{A}_j}(x)$.

Since $x \notin \mathcal{A}_k$ for all $k > j$, then by Corollary 1, the rest of the patches would not change a correct area to an incorrect area. Therefore, we have the final patched neural network $\widehat{f}$ meets specification $\Phi$ on $\cup_{1 \leq l \leq I} \mathcal{A}_l$.

Part 2:

To show that $\widehat{f}$ satisfies $\Phi$ outside of $\mathcal{A}$.

For any $x$ outside the patch area $\cup_{1 \leq l \leq I} \mathcal{A}_l$, we have $x$ lies on a correct linear region (linear region that satisfies the specification $\Phi$). By Theorem 3, we have either $\widehat{f}(x) = f(x)$ or $\widehat{f}(x) \in \Phi_{out}$. Therefore, $\widehat{f}$ satisfies $\Phi$ outside of $\mathcal{A}$. □

**Theorem 4** (Minimum Repair). *For any ReLU DNN $\tilde{f}$, which is linear on a patch region $\mathcal{A}$ and satisfies the specification $\Phi$, there exists a positive number $\Gamma$, such that for all $\gamma \geq \Gamma$,*

$$\max_{x \in X} |\tilde{f}(x) - f(x)| \geq \max_{x \in X} |h_{\mathcal{A}}(x, \gamma)|. \tag{14}$$

*Proof.* We consider the general case where the linear patch function is obtained from Equation (3).

For any DNN $\tilde{f}$, which is linear on patch region $\mathcal{A}$ and satisfies the specification $\Phi$, we have $\max_{x \in \mathcal{A}} |\tilde{f} - f| \geq \max_{x \in \mathcal{A}} |cx + d| = \max_{x \in \mathcal{A}} |h_{\mathcal{A}}(., \gamma)|$ on patch area $\mathcal{A}$ by Equation (3).

Therefore, we only need to show:

$$\max_{x \notin A} |h_{\mathcal{A}}(., \gamma)| \leq \max_{x \in A} |h_{\mathcal{A}}(., \gamma)| \tag{15}$$

Since parameter $\gamma$ controls the slope of $h_{\mathcal{A}}(., \gamma)$ outside of patch area $\mathcal{A}$, a large $\gamma$ means that $h_{\mathcal{A}}(., \gamma)$ will drop to zero quickly outside of $\mathcal{A}$. Therefore, we can choose a large enough $\Gamma$ such that $h_{\mathcal{A}}(., \gamma)$ drops to zero faster than the change of linear patch function $cx + d$.

Therefore, we have that for any $\gamma \geq \Gamma$,

$$\max_{x \notin A} |h_{\mathcal{A}}(., \gamma)| \leq \max_{x \in A} |h_{\mathcal{A}}(., \gamma)| = \max_{x \in X} |h_{\mathcal{A}}(., \gamma)|$$
$$\leq \max_{x \in A} |\tilde{f} - f| \leq \max_{x \in X} |\tilde{f} - f| \tag{16}$$

□

**Theorem 5** (Polynomial-Time Efficiency). *REASSURE terminates in polynomial-time in the size of the neural network and the number of buggy linear regions when $\Phi_{out}$ takes the form of $\{y \mid q_l \leq Py \leq q_u\}$, where $P$ is a full row rank matrix and $-\infty \leq q_l[i] \leq q_u[i] \leq +\infty$ ($q_l[i]$ and $q_u[i]$ are the $i$-th elements of $q_l$ and $q_u$ respectively).*

*Proof.* We consider the affine patch function solved via Equation (10). Suppose $\mathcal{A} = \{x \in X | a_i x \leq b_i, i \in I\}$. For the LPs in Equation (11), $|I|$ is the number of constraints and it is polynomial in the size of the neural network. Thus, REASSURE runs in polynomial time in the size of the neural network.

In addition, since REASSURE computes the support network $g_{\mathcal{A}}$ and affine patch function $p_{\mathcal{A}}$ for each $\mathcal{A}$ one by one (see Algorithm 1), the time complexity of REASSURE is linear in the number of buggy linear regions.

□

## 7.4 ADDITIONAL EXPERIMENT DETAILS

**Details on Feature-Space Repairs:**

For an $R$-layer DNN $f$, we split $f$ into two submodels $f_1$ and $f_2$ according to a hidden layer, say the $j_{th}$ hidden layer, where $f_1$ is the first $j$ layers function, $f_2$ is the last $R - j$ layers function and $f = f_2 \circ f_1$. We note the output space of $f_1$ the feature space. And for any buggy input $\tilde{x}$, we note $f_1(\tilde{x})$ the buggy feature.

Repairing in a feature space is to repair the behavior of $f_2$ on buggy features $\{f_1(\tilde{x}_1), \ldots, f_1(\tilde{x}_L)\}$. Note this will automatically repair the behavior of $f$ on buggy points $\{\tilde{x}_1, \ldots, \tilde{x}_L\}$.

Repairing in a feature space has the benefit of making the repair process more computation-friendly and reducing the parameter overhead of the additional networks, and has the potential to generalize the repair to undetected buggy inputs with similar features. However, it loses the locality guarantee in the input space (but still preserves locality in the feature space).

| | | | REASSURE | | | | | MDNN | | |
|---|---|---|---|---|---|---|---|---|---|---|
| #P | ND($L_\infty$) | ND($L_2$) | NDP($L_\infty$) | NDP($L_2$) | Acc | ND($L_\infty$) | ND($L_2$) | NDP($L_\infty$) | NDP($L_2$) | Acc |
| 1 | **0.0%** | **0.0%** | **9.0%** | **9.1%** | **96.8%** | 8.9% | 8.9% | 7.1% | 6.2% | 87.5% |
| 5 | **0.0%** | **0.0%** | **12.7%** | **12.8%** | **96.8%** | 48.1% | 48.2% | 44.3% | 39.9% | 57.1% |
| 25 | **0.0%** | **0.0%** | **29.9%** | **29.7%** | **96.8%** | 90.4% | 90.6% | 63.7% | 57.8% | 6.7% |
| 50 | **0.0%** | **0.0%** | **42.9%** | **42.6%** | **96.8%** | 92.5% | 92.8% | 82.1% | 72.6% | 4.8% |
| 100 | **0.0%** | **0.0%** | **46.2%** | **45.9%** | **96.8%** | 95.5% | 95.7% | 90.9% | 76.8% | 5.1% |

Table 5: Watermark Removal. The test accuracy of the original DNN is $96.8\%$. #P: number of buggy points to repair; ND($L_\infty$), ND($L_2$): average ($L_\infty$, $L_2$) norm difference on both training data and testing data; NDP($L_\infty$), NDP($L_2$): average ($L_\infty$, $L_2$) norm difference on random sampled points near watermark images; Acc: accuracy on test data.

**Point-wise Repair on ImageNet (Feature-Space Repairs)**

We use AlexNet Krizhevsky et al. (2012) on ImageNet dataset Russakovsky et al. (2015) as the target DNN. The size of image is (224, 224, 3) and the total number of classes for ImageNet is 1000. We slightly modified AlexNet: we only consider 10 output classes that our buggy images may lie on and use a multilayer perceptron with three hidden layers (512, 256, 256 nodes respectively) to mimic the last two layers of AlexNet.

The goal of the repair is to fix the behaviors of the target DNN on buggy inputs, which are found on ImageNet-A Hendrycks et al. (2021). For REASSURE, we construct the patch network starting from the third from the last hidden layer (i.e. repair in a feature space).

The results are shown in Table 4. REASSURE, PRDNN and Fine-Tuning repair all the buggy points while Retrain only repair 96% buggy points in 100 epochs. REASSURE achieves almost zero modification on validation images to the original DNN. In addition, REASSURE preserves the performance of the original DNN.

**Watermark Removal**

We compare REASSURE with MDNN on the watermark removal experiment from their paper. We were not able to run the code provided in the MDNN Github repository, but we were able to run on the target DNN models, watermark images, and MDNN-repaired models in the same repository.

The target DNN is from Goldberger et al. (2020), which is watermarked by the method proposed in Adi et al. (2018) on a set of randomly chosen images $x_i$ with label $f(x_i)$.

The goal is to change the DNN's predictions on all watermarks $x_i$ to any other label $y \neq f(x_i)$ while preserving the DNN's performance on the MNIST test data. For REASSURE, we set the prediction $y = f(x_i) - 1$ if $f(x_i) > 1$, and $y = 10$ otherwise.

The results are shown in Table 5. Both REASSURE and MDNN remove all the watermarks. However, MDNN introduces significant distortion to the target DNN and as a result the test accuracy drops rapidly as the number of repair points increases. In comparison, REASSURE removes all the watermarks with no harm to test accuracy.

**Area Repair: HCAS**

Table 6 is the comparison with PRDNN using the last layer as the repair layer. All other settings are the same as those in Section 5.2.

**Experiment Platform**

All experiments were run on an Intel Core i5 @ 3.4 GHz with 32 GB of memory. We use Gurobi Gurobi Optimization, LLC (2021) to solve the linear programs.

**Size of Neural Networks:**

Point-wise Repairs on MNIST: The DNN model is a multilayer perceptron with ReLU activation functions. It has an input layer with 784 nodes, 2 hidden layers with 256 nodes in each layer, and a final output layer with 10 nodes.

Watermark Removal on MNIST: The DNN model has an input layer with 784 nodes, a single hidden layer with 150 nodes, and a final output layer with 10 nodes.

| | | REASSURE | | | | PRDNN (Last Layer) | | | | |
|---|---|---|---|---|---|---|---|---|---|---|
| #A | $ND(L_\infty)$ | $NDP(L_\infty)$ | NSE | Acc | T | $ND(L_\infty)$ | $NDP(L_\infty)$ | NSE | Acc | T |
| 10 | 2.662e-03% | 1.318e-03% | 0% | 98.1% | 1.0422 | 0.30% | 20.5% | 16% | 71.4% | 2.90+0.100 |
| 20 | 2.918e-03% | 2.2% | 0% | 98.1% | 1.1856 | 0.31% | 46.7% | 66% | 70.5% | 5.81+0.169 |
| 50 | 8.289e-03% | 17.6% | 0% | 98.1% | 1.8174 | 0.31% | 46.7% | 66% | 70.5% | 14.54+0.353 |
| 87 | 0.04% | 45.9% | 0% | 97.8% | 2.4571 | 0.31% | 46.7% | 66% | 70.5% | 25.30+0.467 |

Table 6: Area Repairs on HCAS. We use the the last hidden layer as the repair layer for PRDNN. The test accuracy of the original DNN is $97.9\%$. #A: number of buggy linear regions to repair; $ND(L_\infty)$: average $L_\infty$ norm difference on training data ; $NDP(L_\infty)$: average $L_\infty$ norm difference on random sampled data on input constraints of Specification 1; NSE: $\%$ of correct linear regions that is repaired to incorrect; Acc: accuracy on training data (no testing data available); T: running time in seconds. For PRDNN, the first running time is for enumerating all the vertices of the polytopes and the second is for solving the LP problem in PRDNN.

Area Repair on HCAS: The DNN model has an input layer with 3 nodes, 5 hidden layers with 25 nodes in each hidden layer, and a final output layer with 5 nodes. DNN outputs one of five possible control advisories ('wrong left', 'weak left', 'Clear-of-Conflict', 'weak right' and 'wrong right').

Point-wise Repairs on ImageNet: modified AlexNet Krizhevsky et al. (2012) has 650k neurons, consists of five convolutional layers, some of which are followed by max-pooling layers, and five fully-connected layers.

**Hyperparameters used in Repair:**

We set $\gamma = 0.5$ for Point-wise Repair on MNIST, $\gamma = 0.02$ for Watermark Removal, $\gamma = 1$ for Area Repair: HCAS and $\gamma = 0.0005$ for Point-wise Repair on ImageNet.

We set learning rate to $10^{-3}$ for Retrain in the point-wise repair experiment.

We set learning rate to $10^{-2}$ and momentum to $0.9$ for Fine-Tuning in the point-wise repair experiment.

PRDNN requires specifying a layer for weight modification. We use the first hidden layer as the repair layer, which has the best performance in our experiment settings, unless otherwise specified.

**Specifications in HCAS:**

**Specification 1.** *If the intruder is near and approaching from the left, the network advises "strong right."*

*Input constraints:* $\Phi_{in} = \{(x, y, \psi)|10 \le x \le 5000, 10 \le y \le 5000, -\pi \le \psi \le -1/2\pi\}$. *Output constraint:* $f(x, y, \psi)_4 \ge f(x, y, \psi)_i$ for $i = 0, 1, 2, 3$.

We calculate all the linear regions for $N_{1,4}$ in the area $\Phi_{in}$ of Specification 1 and totally 165 linear regions are found, including 87 buggy linear regions (DNN did not meet the specification) and 78 correct linear regions (DNN meet the specification).

**Specification 2.** *If the intruder is near and approaching from the right, the network advises "strong left."*

*Input constraints:* $\Phi_{in} = \{(x, y, \psi)|10 \le x \le 5000, -5000 \le y \le -10, 1/2\pi \le \psi \le \pi\}$. *Output constraint:* $f(x, y, \psi)_0 \ge f(x, y, \psi)_i$ for $i = 1, 2, 3, 4$.

Also we calculate all the linear regions in the area $\Phi_{in}$ of Specification 2. And 79 correct linear regions are found. We will test if a repair will make those correct linear regions incorrect.

**Point-wise Repairs vs. Area Repairs:**

REASSURE automatically performs area repair on the point-wise repair experiments. This means our area repair method scales well to high-dimensional polytopes (the input dimension of MNIST is 784) whereas PRDNN does not scale beyond 2D linear regions/polytopes.

**Parameter Overhead for REASSURE:**

REASSURE introduces an additional network, patch network, and as a result adds new parameters to the original network. The number of new parameters depends on $|I|$, which is the number of linear

constraints for the H-representation of $\mathcal{A}$. We can remove redundant constraints in this representation in polynomial time to make the additional network smaller. For the area repair experiment on HCAS, the average number of constraints for one linear region is $3.28$ and the average number of new parameters that REASSURE introduces is $66$. As a comparison, the number of parameters in the original network is around $3000$ and PRDNN doubles the number of parameters (as a result of the Decoupled DNN construction) regardless of the number of point-wise or area repairs.

## 7.5 APPLYING REASSURE TO GENERAL CPWL NETWORKS

Recall the result that an $\mathbb{R}^m \to \mathbb{R}$ function is representable by a ReLU DNN *if and only if* it is a continuous piecewise linear (CPWL) function Arora et al. (2016). We use convolutional neural networks as an example to show how REASSURE can be applied to more general CPWL networks. Convolutional neural networks (CNNs) are neural networks with convolution layers and maxpooling layers. For simplicity, we assume the CNNs also use ReLU activation functions (but in general other CPWL activation functions will also work). The convolutional layers can be viewed as special linear layers. The maxpooling layers can be converted to linear operations with ReLU activation functions as follows.

$$\max(x_1, x_2, ..., x_n) = \max(x_1, \max(x_2, x_3, ..., x_n))$$
$$\max(x_i, x_j) = \max(x_i - x_j, 0) + x_j = \sigma(x_i - x_j) + x_j$$

where $\sigma$ is the ReLU activation function. Thus, REASSURE can be used to repair CNNs as well.

