# OpenReview forum: "Sound and Complete Neural Network Repair with Minimality and Locality Guarantees"
_ICLR.cc/2022/Conference — ICLR 2022 Poster_

### Official Review · Reviewer_DRaE · 2021-11-02

**Correctness:** 2
**Technical Novelty And Significance:** 2
**Empirical Novelty And Significance:** 2
**Recommendation:** 5
**Confidence:** 4

**Main Review:**

# Strengths:
1) This paper aims to address an important problem.
2) This paper is well-written and easy to follow.
3) The approach has some novelty.

# Weakness:
1) This paper only focuses on CPWL neural networks, which limit its usefulness.
2) Point-wise repair is a more practical scenario. In real world applications, there are usually a set of buggy inputs such as the misclassified images, texts or audio. Most of existing repair techniques aim to focus on this problem (e.g., retraining). However, this paper requires the linear area for repairing, which is not always provided. Although it provides a way to transform one input point to the linear region in 2.5, there are still some concerns: a) how to guarantee the correctness of the transformed linear regions, the linear region is an over-approximation of the input x?  b) if there are too many failed inputs, then many linear regions will be generated, which may cause the scalability issues. The usage of LP solving may also introduce the scalability issues.  c) In Eq.1, what do you mean $z_j^i <0$? Is it the neuron output of one input $x$?
3) Some description is over-claimed. For example, in Section 1, this paper claimed that, for existing approaches, ''there is no guarantee that the given buggy inputs are fixed and no new bugs are introduced, the Generalization in Section 2.6. It seems that the approach can also not guarantee that no new bugs are introduced  and the claimed generalization, especially for the point-wise repair. Consider the comments (2) above, how to guarantee them? Any support for this claim "the optimization-based approach cannot guarantee removal of the buggy behaviors, and the verification-based approach does not scale beyond networks of a few hundred neurons."?
4) Ensuring the locality is important, but why  Minimality is required? It seems that if we can fix the errors and do not affect the correct inputs, it should be okay. Why Minimality is so important?
5) In Section 3.2, you suppose the linear region is $\{x| a_ix\le b_i, i \in I\}$, is it general for all linear regions?
6) The evaluation is weak. It only compares the proposed method with the existing methods on MNIST for point-wise repair, which may confirm the scalability concerns? Although there are some results on AlexNet, the results are not complete and systematic. For Section 5.1, more discussions are required. For example,  how many failed inputs are fixed by each method? The performance is also not included. How many linear regions are used in different configurations? How is the generalization, i.e., can the repair fix unseen buggy inputs?




**Summary Of The Paper:**

This paper proposed an approach for repairing neural networks that use ReLU activation functions. Different with existing techniques that retrain the model or change the weights, the proposed approach generates a patch in the original model, i.e., add sub-models. The LP solver is used to solve the patch function for a linear region. The experimental  results demonstrated that the approach could outperform existing methods.

**Summary Of The Review:**

Overall, this paper is well-written and interesting. However, considering the limited application, the little discussion on the point-wise repair and the weak experiments, I think the paper should be still improved before it is accepted.

---

> ### Author Response · Authors · 2021-11-13
> **Response to Reviewer DRaE**
>
> We thank the reviewer for the insightful comments and appreciating the novelty of our work. We hope the following response will address the reviewer's concerns.
>
> Response to Weakness in Main Review:
>
> 1. CPWL neural networks represent an important class of neural networks with theoretical guarantees and practical relevance.
> CPWL networks are universal function approximators, i.e. they can approximate any continuous functions.
> Also, many state-of-the-art networks are CPWL networks, such as AlexNet considered in Section 5.3.
> We discuss how to apply REASSURE to general CPWL networks in Appendix 7.5.
> We agree with the reviewer that extending REASSURE to wider classes of activation functions would be an interesting future direction.
>
> 2. a) Linear region is not an over-approximation of the input (see Section 2.2 for the definition of linear region for ReLU DNNs or CPWL functions).
> Geometrically, we can think about linear region as the region where the input resides and the neural network behaves like an affine function in this region.
> We can create an exact representation (as a set of inequalities) of the corresponding linear region for a given input from its neuron activation pattern using Equation (1) in Section 2.5.
>
>     b) The linear regions and the corresponding patch functions can be identified in parallel.
> We do not need to solve an LP to 'generate' the linear regions (Section 2.5 describes how we can obtain a representation of the linear region for a given input).
> We use LPs to compute the patch functions.
> An important result of our work is that we do not need to perform vertex enumeration of the linear region (which is a polytope geometrically) for very general output specification (Theorem 5 with its proof in Appendix 7.3 and details of the LP formulation in Appendix 7.1).
> This allows our approach scale to high-dimensional inputs (e.g. 784-dimensional images for MNIST) which is much better compared to existing approaches especially in the case of area repairs.
>
>     c) $z^i_j$ is the $i$-th neuron's output before activation in the $j$-th hidden layer for input $x$. $z^i_j < 0$ means that the corresponding ReLU unit is not activated (see Section 2.1).
>
> 3. Theorem 3 (Limited Side Effect)  guarantees that no new bugs are introduced by our approach. We provide a proof of this Theorem in Appendix 7.3.
>
>     For the Generalization paragraph in Section 2.6,
> we are saying that
> if there exists another (unknown) buggy input $\widetilde{x}'$ in the same linear region of $\widetilde{x}$, then a good repair method should also fix it. REASSURE does this and in fact fixes all the other infinitely many buggy points that may exist in the same linear region for the given specification.
> In other words, all the points on the linear region are guaranteed to satisfy the correctness specification after REASSURE repairs a buggy point in this region.
> In general, optimization-based approaches (e.g. Retraining which uses stochastic gradient descent) cannot guarantee the removal of buggy behaviors for the given buggy points (see additional results for Retraining on the ImageNet experiment where it failed to repair all the buggy points in 100 epochs).
> Verification-based approach (e.g. using a Satisfiability Modulo Theory solver) is known to not scale
> beyond networks of a few hundred neurons because it encodes the neural network, the buggy points and the correctness specification into a big constraint satisfaction problem that is NP-hard to solve.
> This scalability issue is also stated by the MDNN paper itself.
> We provide a comparison with MDNN in Appendix 7.4 under Watermark Removal.
> In addition, verification-based approach still introduces a global change in the function space and cannot guarantee minimal change to the functional behaviors of the original network (see the large drop on Accuracy in the MDNN column in Table 5 in Appendix 7.4 where we compare REASSURE with MDNN).
> Finally, we would like to emphasize that we are describing verification-based approaches in the context of neural network repair, and not in the context of neural network robustness verification.
> The latter is a completely different problem and certain techniques for solving that problem (which are not applicable here) can scale to large DNNs.
>
> 4. Minimality is to ensure that we can preserve the functional behavior of the original neural network as much as possible.
> In the paper, we note that existing literature on neural network repair considers minimality in the parameter space (i.e. minimal changes to the weights) which does not guarantee minimal changes in the function space even though the latter is what
> we really desire for a good repair method.
>
> 5. Yes, see Section 2.2 for the definition of linear regions.

---

> ### Author Response · Authors · 2021-11-13
> **Response to Reviewer DRaE 2**
>
> Continued from previous response.
>
> 6. For scalability, we have updated the experiment results on AlexNet/ImageNet which now also include the results from Retraining and Fine-Tuning (see general response).
> Similar to the experiments on MNIST, REASSURE significantly outperforms these methods in terms of efficacy of the repair, preserving functional behavior of the original network or its accuracy, and limiting negative side effects.
>
>     In the paragraph right above Section 5.1, we state that
> "REASSURE, MDNN and PRDNN guarantee to repair all the buggy points (linear regions).  Retrain and Fine-Tuning cannot guarantee 100\% efficacy in general and we run them until all the buggy points are repaired."
> For the new results on AlexNet/ImageNet, we highlight that Retrain could only achieve 96\% efficacy in 100 epochs (highest percentage in any of the 100 epochs) and this percentage does not improve even if we train for more epochs (as training has converged).
>
>     Regarding the question on the number of linear regions,
> in the point-wise repair experiments,
> since our method automatically performs area repair on the linear region where each buggy point resides, the number of linear regions is the same as the number of buggy points.
> For the area-repair experiments, the first column in Table 3 (#A) gives the number of buggy linear regions for each experiment. This is stated in the caption of Table 3.
>
>     For generalization, again, REASSURE guarantees fixing any other unseen buggy input $\widetilde{x}'$ in the same linear region that the given buggy input $\widetilde{x}$ resides.

---

### Official Review · Reviewer_Bt2t · 2021-11-02

**Correctness:** 4
**Technical Novelty And Significance:** 3
**Empirical Novelty And Significance:** 2
**Recommendation:** 8
**Confidence:** 4

**Main Review:**

This is a very well written paper which was a pleasure to read.  The paper
draws a clear picture of the state-of-the-art in neural network repair which is
used to  adequately motivate the present contribution. In particular, I think
that the potential side effects that the current state-of-the-art may have
whilst repairing a network, such as introducing a new bug in a previously
non-buggy input, is an important limiting factor of the practical significance
of  neural network repair. It is nice to see an approach towards overcoming
this shortcoming.  I think that the claimed advantages of the approach over
related work are adequately evaluated on an MNIST network and a variant of the
ACAS networks. The scalability of the method is also discussed and additional
experiments on AlexNet are provided. A minor concern that I have regarding the
evaluation is the following.

I think it would be impractical to remove all buggy behaviour (outside of a
test set) from image classifiers. In light of this a metric that is potentially
more important than the ones measured in the experiments is  the robustness of
classifiers w.r.t to input transformations such as noise perturbations. Data
augmentation schemes often aim at improving  precisely this, thus I think that
data augmentation and the present method should be compared on this. So,
although the MNIST dataset was successfully used to showcase the advantages of
REASSURE over related methods, the inclusion of experimental results on
robustness would in my view strengthen the evaluation of the approach.

Some minor comments:

- Page 1-2. loss defined based on -> loss defined on

- Page 3. linear functions -> affine functions.

- Page 6. can solve it is -> can solve it.

- Page 6. with every entry is equal -> where every entry is equal.

- Page 7. REASSURE provide -> REASSURE provides.

- Shouldn't the distance the Theorem 4 be bounded from above?


**Summary Of The Paper:**

The paper develops a method for repairing neural networks with ReLU activation
functions. The method works by augmenting a given network with a patch function that
corrects the output of the network on a given buggy input. This enables the
method to be the only one among related work that is not only sound and
complete but also guarantees (i) minimal distances between  the functions
implemented by the original and repaired networks; (ii) unaltered behaviour of
the repaired network for inputs different than the buggy one.


**Summary Of The Review:**

The paper alleviates some significant limitations in neural network repair in a
method that is adequately evaluated, thereby making for a solid ICLR
contribution.

---

> ### Author Response · Authors · 2021-11-13
> **Response to Reviewer Bt2t**
>
> We thank the reviewer for the insightful comments and appreciating the novelty of our work. We hope the following response will address the reviewer's concerns.
>
> We agree that an evaluation on robustness could be interesting and we have in fact considered it.
> One challenge with such an evaluation is that it depends on the model (and size) of the perturbation.
> Given that our repair is designed to be local with area generalization on the linear region where the buggy point lies, we can expect the repair network to be more robust around a small neighborhood of the buggy point. However, we do not expect increased robustness against other types of perturbations.
> In general, our opinion is that unless the repair technique explicitly takes into account of the model of perturbation, an experiment that shows a repair improves robustness is likely the result of its specific experimental setting and not an inherent guarantee provided by the technique.
>
> We thank the reviewer for pointing out the English issues and typos. We will definitely fix them in the revision.

---

### Official Review · Reviewer_Jhi1 · 2021-11-03

**Correctness:** 4
**Technical Novelty And Significance:** 4
**Empirical Novelty And Significance:** 3
**Recommendation:** 6
**Confidence:** 3

**Main Review:**

The presented techniques are quite interesting and are novel to my knowledge. The presentation of the technique is also clear. I like the running example in section 3. I do have some concerns and questions.
1. Point-wise repair.
To me, the main motivation of point-wise repair is to improve the overall performance (e.g., accuracy) of the network instead of simply making the network perform correctly on a small set of input points. If our goal were the latter, then instead of using the proposed techniques, we could just put the network in a big if-statement which returns the desired output on those points. In other words, in the point-wise repair scenario, we do hope the repair to have side effects, for instance, increase of test accuracy beyond those repaired test images (are the buggy points images in the test set?). It would be beneficial to demonstrate whether this is the case, as right now it looks like the increase in test accuracy results only from fixing the buggy points (with the neural network outputs minimally unchanged).
2. Affine-region repair.
Partitioning the input space into regions where the network is affine and fixing each of those regions sounds feasible for small networks like HCAS. However, it seems challenging to partition the input region into affine sub-regions for large networks or high-dimensional inputs. I do think it is important to show in the main paper 1) the application of the techniques on more complex models; 2) area repairs (for robustness properties for instance) on perception networks.
For HCAS, how are the linear regions represented, boxes?
3. Can the authors comment on the (initial) architectures of the evaluated networks?

Overall, while I think the idea is interesting, more work needs to be done to show the benefit beyond minimal change for point-wise repair and to address the scalability issues.

**Summary Of The Paper:**

This paper presents a new methodology for repairing ReLU neural networks. Concretely, for a single affine region, a patch function is synthesized from an affine function that performs the repair and a support network to make sure the repair only applies to the affine region. Multiple affine regions can be fixed iteratively. The techniques are evaluated on the MNIST and HCAS benchmarks.

**Summary Of The Review:**

Interesting ideas for neural network repair but there are concerns regarding motivation and scalability.

---

> ### Author Response · Authors · 2021-11-13
> **Response to Reviewer Jhi1**
>
> We thank the reviewer for the insightful comments and appreciating the novelty of our work. We hope the following response will address the reviewer's concerns.
>
> Responses to Main Review:
>
> 1. There are two main reasons that 'if-statements' are problematic for the repair problem. First, using 'if statements' breaks the continuous piecewise linear (CPWL) property of ReLU networks. Since a function is representable  by  a  ReLU  DNN if  and  only  if it is  a CPWL function (stated in Section 2.1), using 'if statements' would have no chance of modeling a ground-truth function that is CPWL.
> In addition, losing the CPWL property means we lose the universal approximation guarantee of ReLU networks.
> Second, as stated in Section 2.6 under Generalization, we want the repair of a buggy point to also repair similar buggy points. Consider a buggy point that lies inside some linear region and violates a correctness specification that requires the output to be within some range (a special case of the output specification that we consider in this paper), since the function is affine in this region, we can find infinitely many other buggy points that violate the same specification. An 'if statement' only fixes the single given buggy point whereas our approach will fix all of them (the whole linear region).
>
>     In general, the neural network repair problem (as defined in Section 2.4 and considered in prior literature) aims that fixing all the known (and potentially unknown) buggy points (e.g. because the specification is safety-critical such as in the HCAS experiment in Section 5.2). We list the additional criteria in Section 2.6.
> This is different from learning or generalizing from a set of training examples as repairs are local operations whereas training is global.
> In addition, the original training set is typically much larger than the repair set and may not be available during repair (discussed in Section 1 in the paragraph of Retraining/fine-tuning).
> We consider the MNIST and ImageNet experiments mainly because it allows us to easily set up the pointwise repair problem, i.e. we can treat the misclassified inputs in the test set as buggy points.
> We also directly compare with Retraining and Fine-Tuning in our experiments.
>
> 2.
> Enumerating all the buggy linear regions is indeed  expensive for higher-dimensional inputs, e.g. simply due to the large number of linear regions.
> However, given a buggy point, we can compute the corresponding linear region as described in Section 2.5.
> Then one important result of this paper is that our approach scales well to high-dimensional regions (Theorem 5 and details in Appendix 7.1).
> For example, in the MNIST experiment and the last sentence of the caption for Table 2, we note that REASSURE automatically performs area repairs for the buggy points in a 784 dimensional space.
> Similarly, REASSURE automatically performs area repairs for the ImageNet experiment.
> We consider the HCAS experiment setting for area repairs in Section 5.2 because PRDNN, which is the only other tool that supports area repairs, does not scale beyond two dimensions (because it requires vertex enumeration of the polytope). We noted this in the footnote on page 9.
> Since the input dimension is small for HCAS, we can enumerate all the buggy linear regions to avoid randomness due to sampling in the experiments.
>
>     A linear region is the set of (infinitely many) inputs that correspond to the same activation pattern in a ReLU DNN. Geometrically, this corresponds to a convex polytope which is an intersection of half spaces (see Section 2.2). For HCAS, each linear region is an intersection of some half spaces in 3D, represented as a set of linear inequalities (see Section 2.5).
>
> 3. The architectures of the evaluated networks can be found in Appendix 7.4 under Size of Neural Networks.

---

> > ### Comment · Reviewer_Jhi1 · 2021-11-22
> > **Thank you for your response**
> >
> > I would like to thank the authors for the clarifications. Again I find the techniques quite interesting but I still have some follow-up questions. I'm open to raising my score if they are adequately addressed.
> >
> > 1. I understand that adding if-statements breaks the CPWL property of the neural network. But if this repair is conducted after training (e.g., we are not going to re-train), at that point the CPWL property (which leads to universal approximation guarantee) does not seem crucial. On the other hand, the "if-statement" can also fix affine regions. We can just let the condition in the if-statement be "if the input is in an affine region...".  Of course I'm not claiming the "if-statement" is the best solution for repair. Could the authors speak more to these points? Could the techniques be useful somehow even in combination with re-training (so that the CPWL property is relevant)?
> > 2. I find this response confusing. I am not disagreeing that conducting point-wise repair (i.e., fixing one linear region) is infeasible on networks with high-dimensional inputs. My point is about repairing with respect to a more complex $\Phi_{in}$. There, computing all the  "linear" input regions seems very challenging. I think it's fine that the techniques at their current form only work on low-dimensional input space for non-trivial  $\Phi_{in}$,  but this limitation needs to be discussed in a much less muddy manner.

---

> > > ### Author Response · Authors · 2021-11-23
> > > **Response to the follow-up comments**
> > >
> > > We thank the reviewer for the follow-up comments. We hope our responses below will address the reviewer's remaining concerns.
> > >
> > > 1. By the sentence from the reviewer that "the "if-statement" can also fix affine regions", we believe the reviewer is in agreement with us on the need to fix the whole linear region instead of just the single buggy point. In this case, we want to point out that efficiently computing the patch function for the whole linear region without resorting to enumerating the vertices of a high-dimensional polytope for high-dimensional inputs, is precisely one of the main contributions of our paper.
> > > REASSURE goes one step further and increases slightly the area on which the patch function is applied to so that CPWL is preserved.
> > > Preserving CPWL is important in our opinion because it allows potential downstream operations, in particular any gradient-based computation not limited to retraining, such as gradient-based attribution computation, to be performed on the repaired network the same way as on the original network.
> > > The challenge of preserving CPWL lies in the construction of the support network so that it (1) limits negative side effects due to applying the patch function outside of the linear region and (2) preserves correctness when multiple potentially adjacent regions need to be repaired. Solving this challenge is another main contribution of our paper.
> > >
> > >       We want to additionally note that we already state in Section 3 that the support network $g_A$ behaves like a characteristic function, which is similar to what the reviewer meant by applying an "if-statement" to the linear region. Semantically, both serve the same purpose of deciding if an input is in the linear region. Thus, our disagreement perhaps stemmed more from the different choices of words than from a difference in the technical approaches. We are happy to add a sentence in Section 3 to clarify the meaning of a characteristic function in connection to an "if-statement."
> > >
> > > 2. Since the sentence "I am not disagreeing that conducting point-wise repair (i.e., fixing one linear region) is infeasible on networks with high-dimensional inputs." uses triple negatives, we would like to first confirm if the reviewer is actually saying that point-wise repair is infeasible on networks with high-dimensional inputs.
> > > REASSURE can handle high-dimensional inputs (e.g. 784-dimensional input in the MNIST experiment) for both point-wise and area repair problems, and this is precisely one of our main contributions.
> > > We want to additionally emphasize that computing all the linear regions is not part of our repair process, nor its precondition.
> > > The input specification $\Phi_{in}$, which is a union of some linear regions, is an input to REASSURE.
> > > It can be computed from a set of buggy points via Equation (1).
> > > The set of buggy points may be produced through random testing or by a verification tool.
> > > We enumerated all the linear regions for the HCAS experiment only because the input dimension of the HCAS network is small and we could rule out the randomness that would be otherwise introduced if we sample the input space to look for buggy linear regions.

---

> > > > ### Comment · Reviewer_Jhi1 · 2021-11-23
> > > > **Thank you for the response**
> > > >
> > > > Thank you for the response. I believe my main concerns are adequately addressed and have raised my score to 6. (For 2, sorry for the confusion. I meant to say that I think point-wise repair is feasible on high dimensional input.)

---

### Official Review · Reviewer_fkMv · 2021-11-04

**Correctness:** 3
**Technical Novelty And Significance:** 4
**Empirical Novelty And Significance:** 4
**Recommendation:** 8
**Confidence:** 5

**Main Review:**

Pros:
1. The paper tackles a very relevant but relatively unexplored problem. While there is substantial literature on proving robustness of neural networks, there is only limited work on providing algorithms for provable repair which is as important when considering the deployment of neural networks in the real world. I enjoyed reading the paper, it is well-written. I appreciate that the authors provide intuitive examples to explain the meanings of the different symbols and equations in their algorithmic construction.

2. The technical contributions are novel and solid.

3. The authors provide theoretical results which shows that their algorithms provide stronger guarantees than the state-of-the-art for provable neural network repair.

4. The evaluation is performed on similar benchmarks as prior work and shows significant improvement on obtaining provable repair while minimizing changes to the network.

Cons:
1. The claim about preserving the continuity property needs to be toned down.  The first row in Table 1 says that the proposed repair preserves the continuity property however, the text in Section 3.2 mentions that this is not the case.

2. Scalability to area repairs defined over higher-dimensional polytopes is limited but that is also a limitation of all existing work.


I have a few other comments/questions:

1. The notation in eq. (3) that (c,d) \in {(c,d)...} is weird.

2. For multi-area repair, does different ordering of patch areas yield different repaired networks?.

3. What is the running time for all the tools in Table 2?  Does the timing in Table 3 also include the time to generate vertices?

4. Is it possible to obtain results on a CIFAR10 network in the same setting as the MNIST network or it would be too expensive?

5. What is the value of \gamma used in the experiments? It would be good to show the different metrics vary as a function of /gamma?

6. Similar to DDNN, is extension to arbitrary activations possible for pointwise repair?

**Summary Of The Paper:**

The paper presents a new method for the provable repair of neural networks. Both point-wise and polytope-based repairs are considered. The main idea is to synthesize a patch network and combine it with the buggy network. The patch network consists of two subcomponents: a support network that is only activated for the inputs in the linear region containing buggy inputs and an affine patch network that repairs the network in the buggy region. The experimental evaluation is performed on similar datasets and networks as prior work and shows the effectiveness of the new methodology.

**Summary Of The Review:**

The paper presents a complete and sound methodology for the provable repair of neural networks while minimizing the side effects of the repair. The ideas presented in the paper are new, come with solid theoretical guarantees and advance the state-of-the-art. Based on the contributions, I would recommend acceptance of this work.

---

> ### Author Response · Authors · 2021-11-11
> **Response to Reviewer fkMv**
>
> We thank the reviewer for the insightful comments and appreciating the novelty of our work. We hope the following response will address the reviewer's concerns.
>
> Responses to Cons in the Main Review:
>
> 1. REASSURE does in fact preserve the continuity property of ReLU networks as we claim in Table 1.
> After all, our construction will produce a ReLU network which is continuous piecewise linear (CPWL).
> The sentence "Observe that ..." in Section 3.2 is meant to say that we should not design a support network $g_\mathcal{A}$ that outputs zero for inputs close to the patch linear region $\mathcal{A}$. Because if it does, then the resulting network will be discontinuous and violate the CPWL property. The parameter $\gamma$ in our $g_\mathcal{A}$ controls how fast $g_\mathcal{A}$ goes to 0 outside of $\mathcal{A}$. We will revise this sentence in the paper to make it more clear.
>
> 2. Given the H-representation of a linear region (or equivalently the corresponding convex polytope), which we can compute from a buggy point as described in Section 2.5,
> an important result of this paper is that our approach does not need to enumerate vertices of the polytope and thus scales well to high-dimensional polytopes (see Theorem 5 and details of the LP formulation in Appendix 7.1).
> For example, in the MNIST experiment and in the last sentence of the caption for Table 2, we note that REASSURE automatically performs an area repair for each of the buggy points in a 784 dimensional space.
> Similarly, REASSURE automatically performs area repairs in the ImageNet experiment.
> We consider the HCAS experiment setting for area repairs in Section 5.2 because PRDNN, which is the only other tool that supports area repairs, does not scale beyond two dimensions (because it requires vertex enumeration of the polytope). We noted this in the footnote on page 9.
>
> Responses to questions:
>
> 1. We use $\boldsymbol{c}$ (in bold) to emphasize that it is a matrix.
>
> 2. Yes, ordering matters but only for areas outside the patch areas (the parts where the support network outputs a value between 0 and 1). For the patch areas, the repaired neural networks will have the same behavior regardless of the order.
>
> 3. As described earlier in this response,
> a major feature of REASSURE
> is that it does not
> need to generate or enumerate the vertices of a polytope.
> On the other hand, PRDNN requires vertex enumeration because it essentially reduces the area repair problem to pointwise repairs on each of the vertices of the polytope. Thus, we include the running time for generating/enumerating the vertices for PRDNN in Table 3.
> The runtimes for the pointwise repair experiment are comparable for all the tools. The reasons we did not include them in the submission are: (1) for Retraining and Fine-Tuning, their runtimes depend on the number of training epochs. Right now we stop at the earliest epoch when all the buggy points are repaired. However, this only works for MNIST. For ImageNet, Retraining could only repair 96\% of the buggy points in 100 epochs and increasing the number of epochs further does not increase this percentage; (2) REASSURE is parallelizable -- we can compute the patch for each repair point/area in parallel (and combine them later using the construction described in Section 3.5 to achieve a close-to-linear speedup), but the other methods are not. This means our runtime can be reduced significantly on a many-core machine but we thought doing that might appear unfair.
> As a reference point, running Retrain on the ImageNet dataset for 100 epochs takes 4177.1 seconds with access to one P100 GPU. The runtime for REASSURE on the other hand is 422.8 seconds (roughly 1/10 of that of Retrain) on similar hardware without the need to use a GPU.
>
> 4. The ImageNet experiment in Section 5.3 is meant to evaluate exactly the question of applicability and scalability of our tool on larger DNNs. We will include the additional results for Retraining and Fine-Tuning for this experiment (Table in our general response) in Table 4 in Appendix 7.4.
> Similar to the MNIST experiment, REASSURE demonstrates a significant advantage over existing techniques in terms of efficacy of the repair, preserving functional behavior of the original network or its accuracy, and limiting negative side effects.
> We can expect similar results on CIFAR10.
>
> 5. We will include the specific settings that we used for $\gamma$ in Appendix 7.4 in the revision.
> In general, as long as $\gamma$ is large enough, the results are almost the same.
> In our experience, when the input dimension is small, the rule of thumb is to pick a larger $\gamma$ so that the support function decreases to
> 0 more quickly outside the patch area.
> Identifying a lower bound for $\gamma$ is a subject of future work as stated in Section 6.
>
> 6. The current framework works for CPWL activation functions in general. Generalizing REASSURE to wider classes of activation functions is definitely a subject of future work.

---

> > ### Comment · Reviewer_fkMv · 2021-11-28
> > **Claims still not clear**
> >
> > Dear Authors,
> >
> > Thanks for the detailed response, I am still positive about the paper as it advances the state-of-the-art but some of the claims in the response follow the same pattern I mentioned in the **cons** section. Theorem 5 makes the assumption that P is a full-row rank matrix which is an easy case to handle. Therefore the complexity of the general case may remain. Further, in Table 2, the authors added a sentence that they do the area repair as well which is a natural artifact of their approach, they should consider adding the exact area repair specification that they solve and whether that is useful in any way.
> >
> > Further, I still do not get the claim about CPWL, can I choose a value of gamma that will break the CPWL property? If yes, then this should be explicitly mentioned.
> >
> > I would encourage the authors to provide more details and not to provide misimpression about the work.

---

> > > ### Author Response · Authors · 2021-11-28
> > > **Response to the follow-up comments**
> > >
> > > We thank the reviewer for the follow-up comments. We hope our responses below will address the reviewer's remaining concerns.
> > >
> > > 1. Assumption on $P$ being a full-row rank matrix.
> > >
> > >     REASSURE does not rely on this assumption to work but will require vertex enumeration in the general case as the reviewer pointed out (see Equation (6) and the paragraph right below it). On the other hand, identifying this technical condition and thus circumventing vertex enumeration for much better efficiency is exactly a main contribution of this paper. Also note that the same technical condition does not help the other approaches such as PRDNN.
> > >
> > >     For the condition that $P$ being a full-row rank matrix itself, it is the same as saying that the number of constraints is less than or equal to the number of outputs involved in the specification. For real-world applications, $\Phi_{out}$ is typically not too complex to describe and thus would not have many constraints. In fact, for all the experiments we conducted in our paper, including image classification (MNIST and ImageNet), watermark removal (in Appendix 7.4 where we compare REASSURE with MDNN) and the HCAS specifications, they meet the condition that $P$ is a full-row rank matrix.
> > >
> > > 2. The exact area repair specification for Table 2.
> > >
> > >     In Table 2, we italicize the sentence "Note that REASSURE automatically performs area repairs on 784-dimensional inputs." in the revision to highlight that REASSURE can scale well to high-dimensional inputs for area repairs since we had to pick HCAS which has only 3 inputs in order to compare with the prior art in area repairs.
> > > For this experiment, the specification $\Phi_{in}$ is generated by applying Lemma 1 to every buggy input.
> > > We did not give the exact specification of $\Phi_{in}$ in text because it is a linear region on the MNIST image space which has 784 dimensions and it will take too much space to include all the constraints in the paper.
> > > As for $\Phi_{out}$, Example 1 in Section 2.3 gives the exact specification for this case. We would like to again note that this specification meets the technical condition of $P$ being a full-row rank matrix. Thus, REASSURE can perform the repair efficiently without needing to enumerate the vertices of 784-dimensional polytopes.
> > >
> > > 3. Is there a value of gamma that will break the CPWL property?
> > >
> > >     No, there is not. For any gamma, we can guarantee the CPWL property since the repaired network remains a ReLU network. As we described in Section 3.2, the value of gamma controls how quickly the support network $g_A(x, \gamma)$ goes to zero outside of $A$, i.e. the slope of the support network (see the top-right illustration of a support network in Figure 2). The larger the $\gamma$ is, the steeper the slope and thus the more quickly the support network goes to zero. In other words, $\gamma$ determines how much the patch function will affect the areas outside of $A$. Due to the page limit, we have moved this explanation to the Appendix (see the paragraph right after Equation (15)). We will move it back to the main paper in the revision given this feedback.

---

### Author Response · Authors · 2021-11-11
**Additional results for the Point-wise Repair experiment on ImageNet**

We plan to use normalized norm difference for all tables in the revision for ease of comparison. The following table includes the additional results for Retraining and Fine-Tuning for the Point-wise Repair experiment on ImageNet.

| | Reassure(feature space) | Retrain(Requires Training Data) | Fine-Tuning | PRDNN |
|---| --- | --- |--- |--- |
|#P|ND($L_{\infty}$) & ND($L_2$) & Acc|ND($L_{\infty}$) & ND($L_2$) & Acc|ND($L_{\infty}$) & ND($L_2$) & Acc|ND($L_{\infty}$) & ND($L_2$) & Acc|
|10| __0.15\%__ & __0.13\%__ & __82.5\%__ | 43.43\% & 36.04\% & 80.1\% |29.45\% & 25.27\% & 77.9\% |22.93\% & 21.13\% & 82.1\% |
|20| __0.12\%__ & __0.11\%__ & 81.3\% | 42.78\% & 35.69\% & __82.9\%__ |69.16\% & 57.47\% & 68.5\% |21.91\% & 20.03\% & 80.1\% |
|50| __0.79\%__ & __0.70\%__ & 81.3\% | 50.23\%* & 42.86\%* & __82.1\%*__ |76.69\% & 63.46\% & 66.9\% |  30.96\% & 26.94\% & 68.9\% |

Point-wise Repairs on ImageNet. PRDNN uses parameters in the last layer for repair. The test accuracy for the original DNN is 83.1%. \#P: number of buggy points to repair.
ND($L_\infty$), ND($L_2$): average ($L_\infty$, $L_2$) normalized norm difference on validation data.
Acc: accuracy on validation data. * means Retrain only repair 96\% buggy points in 100 epochs.

---

### Author Response · Authors · 2021-11-15
**Summary of the changes we made in the revision**

Below is a summary of the changes we made in the revision:
1. Revised some of the sentences to improve clarity and fixed the typos pointed out by the reviewers.
2. Included the additional results for Retraining and Fine-Tuning for the point-wise repair experiment on ImageNet in Appendix 7.4.
3. Included the specific settings that we used for $\gamma$ in Appendix 7.4.
4. Highlight that REASSURE automatically performs area repairs for the point-wise repair experiments in the caption of Table 2;
5. Normalize the ND and NDP numbers in the revision for better interpretability of the results.

---

### Decision · Program_Chairs · 2022-01-20

**Decision:**

Accept (Poster)

**Comment:**

Reviewers were almost unanimous in favor of this paper, with scores of 5,8,6,8.
I think it's a neat idea and am inclined to accept despite some issues w/ motivation / scalability.
Science proceeds in increments, and it's OK to propose something with scalability issues that someone else later tries to fix, etc.